# MolChord: Structure–Sequence Alignment for Protein-Guided Drug Design

## Abstract

Structure-based drug design (SBDD), which maps target proteins to candidate molecular ligands, is a fundamental task in drug discovery. Effectively aligning protein structural representations with molecular representations, and ensuring alignment between generated drugs and their pharmacological properties, remains a critical challenge. To address these challenges, we propose MolChord, which integrates two key techniques: (1) to align protein and molecule structures with their textual descriptions and sequential representations (e.g., FASTA for proteins and SMILES for molecules), we leverage NatureLM, an autoregressive model unifying text, small molecules, and proteins, as the molecule generator, alongside a diffusion-based structure encoder; and (2) to guide molecules toward desired properties, we curate a property-aware dataset by integrating preference data and refine the alignment process using Direct Preference Optimization (DPO). Experimental results on CrossDocked2020 demonstrate that our approach achieves state-of-the-art performance on key evaluation metrics, highlighting its potential as a practical tool for SBDD.

## 1 Introduction

Drug discovery is a long and costly process, often spanning over a decade and requiring billions of dollars in investment (Paul et al., 2010; DiMasi et al., 2016). The chemical space is estimated to contain up to $10^{60}$ synthetically accessible molecules (Polishchuk et al., 2013), making it infeasible explore all possibilities. Structure-based drug design (SBDD) has emerged as a transformative approach in drug discovery (Anderson, 2003; Batool et al., 2019; Schneider et al., 2020), leveraging the structure of biological targets to rationally design drug compounds using computational techniques like molecular docking. Recent advances in artificial intelligence (AI) have further enhanced SBDD (Luo et al., 2021; Peng et al., 2022; Guan et al., 2023a), with typical frameworks employing protein encoders to transform protein structures into high-dimensional representations and generators to map these representations back into the chemical space (Wu et al., 2024; Feng et al., 2024), either as 3D molecular structures or chemical descriptors. These advancements significantly improve the efficiency and accuracy of drug design.

Despite these advancements, aligning protein representations with molecular representations remains a challenge for AI-based SBDD, mainly due to the limited number of high-quality protein–ligand pairs (Feng et al., 2023; Gao et al., 2023). Furthermore, ensuring that generated compounds are aligned with desired drug properties presents another critical issue. However, generating large-scale, high-quality protein–ligand data is prohibitively expensive and time-consuming (Davies et al., 2006; Nakata et al., 2023). Instead of solely relying on building more protein–ligand datasets with structural information, we propose exploring novel approaches to improve the alignment between structure encoders and chemical generators.

A promising trend in research is the development of unified scientific entity generators, such as MolXPT (Liu et al., 2023) (text, small molecule), LucaOne (He et al., 2025) (protein, DNA, RNA), and NatureLM (Xia et al., 2025) (text, molecule, DNA, RNA, protein, material), which are designed to jointly model diverse biological and chemical sequences within a unified representational space. By adopting such a unified generator in AI-based SBDD models, alignment between structure encoders and molecule generators can be enhanced through tasks like protein-to-text and protein-to-FASTA transformations, whose data are substantially larger in scale compared to protein–ligand

pairs. These tasks facilitate more effective alignment by enabling encoders and generators to learn across multiple modalities.

In this work, we introduce MOLCHORD, a four-billion-parameter framework with enhanced alignment between the structure encoder and sequence generator. The structure encoder is a diffusion-based model pre-trained to capture geometric and structural features (residue-level for proteins and atom-level for molecules). For the generator, we implement a variant of NatureLM (Xia et al., 2025), an autoregressive sequence generator capable of handling protein FASTA sequences, molecular SMILES, and text representations. Our training process consists of three stages to achieve robust alignment. First, the structure encoder and sequence generator are connected via a lightweight adapter, pre-trained on five structure-to-sequence tasks: protein-to-FASTA, protein-to-text, molecule-to-SMILES, molecule-to-text, and complex-to-FASTA/SMILES. This pre-training establishes a shared representational space across proteins and molecules. Next, we perform supervised fine-tuning on pocket–ligand complexes to anchor the model with biological evidence. Finally, we apply Direct Preference Optimization (DPO) to a curated subset of CrossDocked2020 (Francoeur et al., 2020), which provides reliable preference signals and broad protein coverage. This curation enables reinforcement learning to improve binding affinity while maintaining validity, synthesizability, and diversity. Through this staged design, MOLCHORD achieves scalable and effective protein–ligand alignment, yielding a unified foundation model that advances the practicality of SBDD.

We systematically evaluate MOLCHORD on CrossDocked2020 (Francoeur et al., 2020), the widely used dataset for SBDD. MOLCHORD consistently outperforms strong baselines on affinity-related proxies while preserving synthesizability (SA), quantitative estimate of drug-likeness (QED), and scaffold diversity. The gains are more pronounced under limited paired supervision and on held-out targets, indicating robust cross-modal alignment rather than overfitting to heuristics. Ablations show that both the diffusion-pretrained structure encoder and DPO fine-tuning are necessary; removing either degrades the affinity–drug-likeness trade-off. These results validate our design choice of coupling diffusion-based encoding with autoregressive generation via a lightweight sequential/textual adapter.

Our contribution can be summarized as follows:

- We propose MOLCHORD, a unified framework that leverages diffusion to capture protein structure and autoregression for SMILES generation, aligning protein, molecule, and text representations in target-aware molecular design.

- We curate a property-aware dataset for reinforcement learning and apply Direct Preference Optimization (DPO) to refine alignment, improving binding affinity while preserving other molecular properties.

- Experimental results on CrossDocked2020 datasets demonstrate that MOLCHORD achieves state-of-the-art performance on key evaluation metrics, underscoring its potential as a practical tool for structure-based drug design.

## 2 RELATED WORKS

**Structure-based Drug Design** Structure-based drug design aims to design ligands conditioned on protein structures or sequences. Early representative works include liGAN (Ragoza et al., 2022), which voxelizes protein–ligand complexes into atomic density grids within a conditional VAE framework, and GraphBP (Liu et al., 2022), which generates ligands through graph-based placement in 3D binding pockets. Building on these foundations, recent work can be broadly categorized into three families: diffusion-based, flow-based, and autoregressive approaches. Diffusion-based methods model protein–ligand distributions in continuous 3D space, including DiffSBDD (Schneuing et al., 2024), TargetDiff (Guan et al., 2023a) with SE(3)-equivariant denoising, and DecompDiff (Guan et al., 2023b), which incorporates functional-region decomposition to improve validity and synthesizability. Flow-based approaches parameterize generation in continuous latent space, such as FlowSBDD (Zhang et al., 2024) and MolForm (Huang & Zhang, 2025), which leverage rectified or multimodal flow matching for molecular design. Autoregressive (AR) models formulate ligand design as conditional sequence generation. Early examples include AR (Luo et al., 2021), Pocket2Mol (Peng et al., 2022), and ResGen (Zhang et al., 2023a), which autoregressively generate ligands conditioned on binding pockets. Among them, ResGen leverages residue-level encoding,

while Pocket2Mol operates at the atom level. More recent developments adopt tokenization of structural inputs: XYZ-Transformer (Flam-Shepherd & Aspuru-Guzik, 2023) and BindGPT (Zholus et al., 2025) directly treat 3D coordinates as tokens for autoregressive modeling. In addition, several works incorporate an explicit structure encoder to enrich conditional signals, including Tam-Gen (Wu et al., 2024), 3D-SMILES-GPT (Wang et al., 2025), and Lingo3DMol (Feng et al., 2024). This line of work is most closely related to our approach, yet our method distinguishes itself by scaling model capacity and introducing principled cross-modal alignment.

**Reinforcement Learning** Likelihood training is standard in generative modeling, yet often misaligned with user objectives, motivating reinforcement learning for alignment. In particular, reinforcement learning from human feedback (RLHF) (Ziegler et al., 2019; Ouyang et al., 2022) has proven effective in steering LLM toward human intent. More recently, Direct Preference Optimization (DPO) (Rafailov et al., 2023) has emerged as a lightweight alternative that bypasses explicit reward modeling by directly optimizing on preference pairs, achieving results comparable to RLHF while being simpler and more stable to train. Recently, several studies have explored reinforcement learning in structure-based drug design. BindGPT (Zholus et al., 2025) and 3DMolFormer (Hu et al., 2025) integrate RL objectives to enhance binding affinity, while DecompDPO (Cheng et al., 2024) introduces a decomposition-based alignment scheme to better guide optimization. Other approaches have incorporated preference-based learning into SBDD: MolForm (Huang & Zhang, 2025) applies Direct Preference Optimization (DPO) to improve docking affinity, and AliDiff (Gu et al., 2024) proposes Exact Energy Preference Optimization ($E^2PO$) with additional regularization. Despite these advances, BindGPT, 3DMolFormer, and DecompDPO tend to improve affinity at the cost of molecular diversity, whereas preference-based approaches like MolForm and AliDiff remain heavily tied to docking scores, often degrading key properties such as QED and synthesizability. These limitations point to the need for higher-quality preference data and more principled optimization objectives.

## 3 METHOD

In this section, we present MOLCHORD, our framework for structure-based drug design. We begin with the problem definition in Section 3.1, and then describe the overall architecture in Section 3.2. The training strategy is introduced in Section 3.3.

### 3.1 PROBLEM DEFINITION

SBDD can be formulated as conditional molecule generation given a protein pocket. Let $P^{\text{prot}} = \{(\mathbf{x}_i^{\text{res}}, \mathbf{a}_i)\}_{i=1}^{N_{\text{res}}}$ denote a protein, where $\mathbf{x}_i^{\text{res}} \in \mathbb{R}^3$ is the 3D coordinate of the $\alpha$-carbon atom of the $i$-th residue, and $\mathbf{a}_i$ denotes residue-level annotations such as amino acid type, chain identity, and residue index. A binding pocket $P^{\text{pock}} \subset P^{\text{prot}}$ is defined as the subset of residues surrounding the active site. The goal of SBDD is to generate a ligand $M$ that can bind to $P^{\text{pock}}$. In this work, we focus on designing compounds in chemical space and let $M$ denote the SMILES sequence $M = (m_1, m_2, \ldots, m_{|M|})$ with $s_i$ representing the $i$-th token in the SMILES sequence.

### 3.2 MODEL ARCHITECTURE

As illustrated in Figure 1, the architecture consists of three main modules: a structure encoder (`Encoder`) that encodes structures of molecule, protein and complex; a sequence generator (`Generator`) responsible for generating SMILES and related sequences; an adapter (`Adapter`) with an auxiliary variational autoencoder (`VAE`) to align `Encoder` and `Generator`. Our model has 4.2B parameters in total. For each reference, denote the embedding layer of the `Generator` as `embed`, which maps discrete sequence tokens into hidden representations.

**Structure Encoder** The `Encoder` is pre-trained using a diffusion-based objective and is capable of processing protein structures, molecular structures, and protein–molecule complex structures within a single model. The input is defined as $X = \{(\mathbf{x}_i, \mathbf{a}_i)\}_{i=1}^{N_{\text{tok}}}$, where $\mathbf{x}_i$ and $\mathbf{a}_i$ denote the coordinates and the annotation of the $i$-th element in $X$. Protein structures are represented at the residue level, molecular structures are represented at the atom level, and complex structures are represented with a combination of residues for the protein component and atoms for the molecular component.

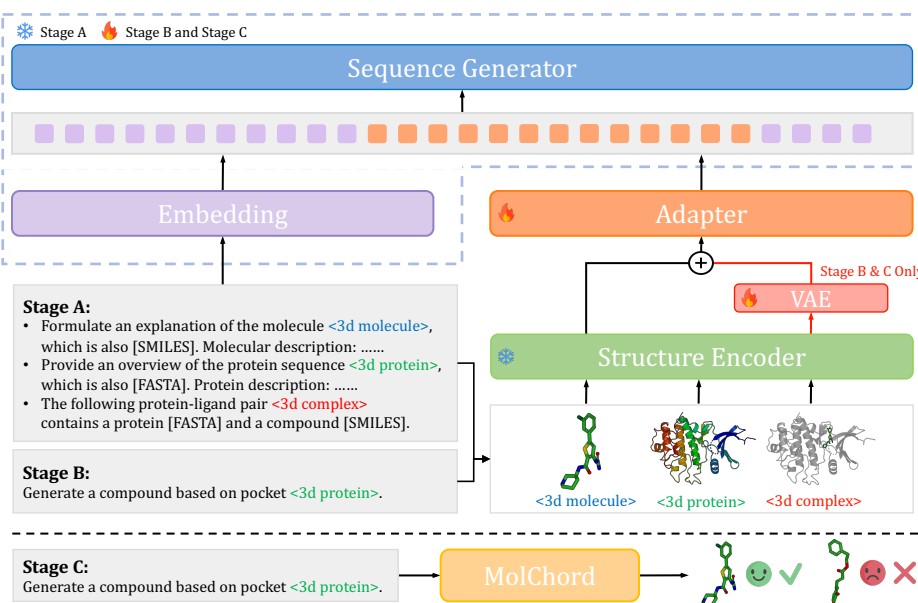

Figure 1: Overview of MOLCHORD. For each input, unmarked text tokens are embedded by the language model, while color-marked entities (⟨3d molecule⟩, ⟨3d protein⟩, or ⟨3d complex⟩) are processed by the Encoder. In Stage B, protein–ligand complexes are further processed through a VAE to perturb protein features, and only pocket features are injected into the language model. The bottom panel illustrates Stage C, where Direct Preference Optimization (DPO) is applied.

The architecture of the Encoder primarily follows the Elucidated Diffusion Model (EDM) (Karras et al., 2022), a variant of the Transformer architecture that incorporates geometric information. Encoder is pre-trained on AlphaFoldDB (Varadi et al., 2024) and PDB (Berman et al., 2000). Additional details about pre-training setups can be found in Appendix A.1. By using the Encoder, for each $(\mathbf{x}_i, \mathbf{a}_i) \in X$, we can obtain a contextual representation $\texttt{Encoder}(X)$.

**Generator** Following Xia et al. (2025), the Generator is a language model pretrained on molecule SMILES, protein FASTA sequences, and textual annotations by using next token prediction. Further details are provided in Appendix A.2. The pretraining of the Encoder and the Generator is conducted independently.

**Align the Encoder and Generator** Given a 3D structure input $X$ and its corresponding annotation, we demonstrate how the Encoder and Generator are jointly utilized. Together, they form an interleaved sequence like:

$$I = (t_1, t_2, \cdots, t_m, (\mathbf{x}_1, \mathbf{a}_1), (\mathbf{x}_2, \mathbf{a}_2), \cdots, (\mathbf{x}_i, \mathbf{a}_{N_{\text{tok}}}), t_{m+1}, t_{m+2}, \cdots, t_n), \quad (1)$$

where $t_i$ represents tokens such as text, SMILES, or FASTA.

For instance, see the first input of Stage A in Figure 1, where the prefix $(t_1, \ldots, t_m)$ corresponds to the text *"Formulate an explanation of the molecule"*, the suffix $(t_{m+1}, \ldots, t_n)$ corresponds to *"which is also [SMILES]. Molecular description: ..."*, and the placeholder ⟨3d molecule⟩ is expanded into $(\mathbf{x}_1, \mathbf{a}_1), \ldots, (\mathbf{x}_{N_{\text{tok}}}, \mathbf{a}_{N_{\text{tok}}})$, which together constitute the 3D input $X$ of the molecule in $I$.

The 3D input $X$ in $I$ is first processed as

$$\mathbf{U} = \texttt{Adapter}(\texttt{Encoder}(X)), \quad (2)$$

where each $(\mathbf{x}_i, \mathbf{a}_i)$ in $X$ is processed into $\mathbf{u}_i$, a high-dimensional representations and $U = (\mathbf{u}_1, \mathbf{u}_2, \cdots)$. The $t_i$ in $I$ is mapped by embed layer and obtain $e_i = \texttt{embed}(t_i)$. By this way,

all elements in $I$ are mapped as

$$I_{\text{emb}} = (e_1, \cdots, e_m, \mathbf{u}_1, \cdots, \mathbf{u}_N, e_{m+1}, \cdots, e_n). \tag{3}$$

The embedded sequence $I_{\text{emb}}$ is then fed into the embedding layer of `Generator` to perform the generation task. This formulation unifies structural and textual tokens into a single embedding sequence, allowing the `Generator` to attend jointly over structural representations and symbolic annotations.

### 3.3 TRAINING STRATEGY

We adopt a three-stage training strategy. In Stage A, we train only the parameters of the `Adapter` to align the `Encoder` with the `Generator`. In Stage B, we perform supervised fine-tuning on protein-ligand data to enhance the protein-to-ligand generation capability. Finally, in Stage C, we apply direct preference optimization (DPO) to align the model with key preferences essential for SBDD.

Denote the dataset of stage A as $\mathcal{D}_{\text{A}}$, which consists of the following datasets for alignment: (i) 676K protein structures paired with FASTA sequences and functional annotations, collected from multiple sources including PDB (Berman et al., 2000) and SwissProt (Boutet et al., 2007); (ii) 316K small molecules paired with SMILES and textual descriptions, collected from Uni-Mol (Zhou et al., 2023); and (iii) 94K protein–ligand complexes annotated with both 3D coordinates, obtained from PDB (Berman et al., 2000). All datasets are processed into interleaved sequences (see Eqn. (1)).

For Stages B and C, we exclusively use protein–ligand complexes from CrossDocked2020 (Francoeur et al., 2020), which are subsequently divided into two disjoint datasets: $\mathcal{D}_{\text{B}}$ and $\mathcal{D}_{\text{C}}$. If a protein is associated with $> 2$ molecules, it is assigned to $\mathcal{D}_{\text{B}}$; otherwise it is assigned to $\mathcal{D}_{\text{C}}$. The intuition behind this strategy is two-fold: (i) In large language model (LLM) training, it is typical to maintain disjoint datasets for supervised fine-tuning (SFT) and reinforcement learning (or decision preference optimization), as these stages have distinct objectives; (ii) for our task, if a protein pocket is associated with only one or two ligands, the `Generator` is less likely to produce diverse molecules, making it less effective for alignment purposes. Assigning such pairs to $\mathcal{D}_{\text{C}}$ ensures a focus on alignment, while $\mathcal{D}_{\text{B}}$ benefits from more diverse multi-ligand associations.

**Stage A:** We freeze the `Encoder` and `Generator`, training only the `Adapter` that maps structural features to the embedding space of the `Generator`. This is achieved through next-token prediction:

$$\mathcal{L}_{\text{alignment}} = -\frac{1}{|\mathcal{D}_A|} \sum_{I \in \mathcal{D}_A} \sum_{i=\texttt{fid}(I)}^{|I|} \log P(I_i | I_{<i}); \tag{4}$$

where $\texttt{fid}(I)$ denotes the first index following the 3D structure element (i.e, the index of $e_{m+1}$ in the $I$ of Eqn. (1)), and $|I|$ is the sequence length of $I$.

**Stage B:** The model is then fine-tuned on the protein-ligand dataset. We adopt a variational autoencoder (VAE)-based approach in this stage to increase the diversity of the generated molecules. During training, a controlled noise term is injected into the `Adapter` as follows:

$$\begin{aligned} (\boldsymbol{\mu}, \boldsymbol{\Sigma}) &= \texttt{VAE}(\texttt{Encoder}(P^{\texttt{prot}}, M^{\texttt{ref}})), \\ \mathbf{u} &= \texttt{Adapter}\left(\texttt{Encoder}(P^{\texttt{prot}}) + \boldsymbol{\epsilon}\right). \end{aligned} \tag{5}$$

In Eqn. (5), (i) `VAE` is a feed-forward layer that outputs the mean $\boldsymbol{\mu}$ and variance $\boldsymbol{\Sigma}$; (ii) $\boldsymbol{\epsilon}$ is sampled from the Gaussian distribution $\mathcal{N}(\boldsymbol{\mu}, \boldsymbol{\Sigma})$. During inference, $\boldsymbol{\epsilon}$ is sampled from standard Gaussian distribution $\mathcal{N}(\mathbf{0}, \boldsymbol{I})$.

The output $\mathbf{u}$ is then used to construct a new interleaved sequence $I$ in Eqn. (3). During Stage B, the `Encoder` and `Adapter` process the entire protein structure, while only the features corresponding to the binding pocket are injected into the `Generator`.

The overall training objective for Stage B is defined as:

$$\mathcal{L}_{\text{SFT}} = -\frac{1}{|\mathcal{D}_{\text{B}}|} \sum_{I \in \mathcal{D}_{\text{B}}} \sum_{i=\texttt{ind}(I)}^{|I|} \log P(I_i | I_{<i}) + \beta_{\text{vae}} D_{\text{KL}}[p(\boldsymbol{\epsilon}) \| \mathcal{N}(\mathbf{0}, \boldsymbol{I})], \tag{6}$$

where $\beta_{\text{vae}} > 0$ is the hyperparameter.

**Stage C:** The core aspect of DPO is constructing the preference data. For each pocket in $\mathcal{D}_{\text{C}}$, we sample 100 candidate molecules using the checkpoint from Stage B with the lowest validation loss. A pocket is retained for further processing if the diversity among these 100 candidates exceeds 0.8. The diversity is measured as $1 - \sum_{i=1}^{100} \sum_{j=i+1}^{100} \texttt{fingerprint\_similarity}(M_i, M_j)/Z$ where $Z$ is the normalization factor. By this way, about 1K protein pockets are selected, denoted as $\mathcal{D}_{\text{DPO}}$. The reward for each sampled molecule $M$ is then defined as:

$$R(M, P^{\text{pock}}) = -\left(S_{\text{Vina}}(M, P^{\text{pock}}) + \lambda \cdot \max(0, \ \#\texttt{fused\_ring}(M) - 2)\right) \quad (7)$$

where $S_{\text{Vina}}$ is the docking score computed by AutoDock Vina (a lower docking score indicates better binding affinity), $\lambda$ denotes fused ring penalty, and $\#\texttt{fused\_ring}(M)$ represents the number of fused rings in molecule $M$ (a lower fused ring count may suggest that $M$ is easier to synthesize and have reduced toxicity). This quantity is strongly correlated with the molecule's quantitative estimate of drug-likeness (QED) and its synthetic accessibility. The molecules with the highest and lowest rewards are denoted as $M^+$ and $M^-$ respectively. The reward function is defined as follows:

$$\mathcal{L}_{\text{DPO}} = -\log \sigma \left( \beta_{\text{DPO}} \left[ \log \frac{\pi(M^+ \mid P^{\text{pock}})}{\pi_{\text{ref}}(M^+ \mid P^{\text{pock}})} - \log \frac{\pi(M^- \mid P^{\text{pock}})}{\pi_{\text{ref}}(M^- \mid P^{\text{pock}})} \right] \right), \quad (8)$$

where $\pi_{\text{ref}}$ is the frozen model from Stage B and $\beta_{\text{DPO}}$ controls preference sharpness. Note that the variational encoder loss is also included in Stage C.

# 4 EXPERIMENTS

## 4.1 EXPERIMENTAL SETUP

**Dataset** To align with prior work (Luo et al., 2021; Peng et al., 2022), we use CrossDocked2020 (Francoeur et al., 2020) to fine-tune and evaluate our model. We adopt the preprocessing and splitting procedure described in (Luo et al., 2021). Starting from 22.5M docked protein–ligand complexes, we keep only those with RMSD to the ground truth below 1Å and with protein sequence identity under 30%. This results in a curated set of 100,000 complexes for training and 100 proteins reserved for testing. The training set is further divided for SFT and DPO (see Section 3.3).

**Baselines** We benchmark MOLCHORD against a range of representative baselines for target-aware molecular generation: prior structure-based models (**liGAN** (Ragoza et al., 2022), **GraphBP** (Liu et al., 2022)); autoregressive approaches (**AR** (Luo et al., 2021), **Pocket2Mol** (Peng et al., 2022), **TamGen** (Wu et al., 2024)); diffusion-based methods (**TargetDiff** (Guan et al., 2023a), **DecompDiff** (Guan et al., 2023b)); the BFN-based **MolCRAFT** (Qu et al., 2024); and the flow-based **FlowSBDD** (Zhang et al., 2024). Together, these baselines span diverse methodological families and provide a balanced foundation for evaluating the effectiveness of MOLCHORD.

**Evaluation** To provide a comprehensive assessment of generated molecules in drug design applications, we consider the following evaluation metrics: (1) **Vina Dock**, denoting the binding affinity score estimated via re-docking; (2) **High Affinity**, measuring for each pocket the fraction of generated molecules that achieve Vina Dock scores no worse than the corresponding test-set ligands;(3) **QED** (Quantitative Estimate of Drug-likeness) (Bickerton et al., 2012) ;(4) **SA** (Synthetic Accessibility) (Ertl & Schuffenhauer, 2009; You et al., 2018) ;(5) **Diversity**, computed as the average pairwise Tanimoto similarity among generated molecules within each pocket;(6) **Success Rate**, representing the fraction of molecules that are drug-like, synthesizable, and high-affinity binders, is computed following (Long et al., 2022) and (Guan et al., 2023b) as the proportion of molecules with QED > 0.25, SA > 0.59, and Vina Dock < −8.18. To evaluate binding affinity to the target, we use AutoDock Vina (Eberhardt et al., 2021), adopting the evaluation protocol described by (Guan et al., 2023a). For each protein pocket, we evaluate 100 generated molecules.

## 4.2 MAIN RESULTS

Table 1 summarizes the performance of MOLCHORD and its RL variants, including MOLCHORD-RL and MOLCHORD-RL$^{\text{dock}}$, where the latter denotes the model optimized with DPO solely for

Table 1: Summary of molecular properties between MOLCHORD and other baseline methods for pocket-aware drug design. (↑) / (↓) indicates larger / smaller is better. Top-2 results are marked in **bold** and underline, respectively. A more comprehensive version of this table is provided in Table 11 in Appendix.

| Methods | Vina Dock (↓) | High Affinity (↑) | QED (↑) | SA (↑) | Diversity (↑) | Success Rate (↑) |
|---------|---------------|-------------------|---------|--------|---------------|------------------|
| Reference | -7.45 | - | 0.48 | 0.73 | - | 25.0% |
| LiGAN | -6.33 | 21.1% | 0.39 | 0.59 | 0.66 | 3.9% |
| GraphBP | -4.80 | 14.2% | 0.43 | 0.49 | **0.79** | 0.1% |
| AR | -6.75 | 37.9% | 0.51 | 0.63 | 0.70 | 7.1% |
| Pocket2Mol | -7.15 | 48.4% | **0.56** | 0.74 | 0.69 | 24.4% |
| TamGen | -7.48 | 52.6% | **0.56** | 0.77 | 0.75 | 32.4% |
| TargetDiff | -7.80 | 58.1% | 0.48 | 0.58 | 0.72 | 10.5% |
| DecompDiff | -8.39 | 64.4% | 0.45 | 0.61 | 0.68 | 24.5% |
| MolCRAFT | -7.92 | 59.1% | 0.50 | 0.69 | 0.72 | 26.8% |
| FlowSBDD | -8.50 | 63.4% | 0.47 | 0.51 | 0.75 | - |
| MOLCHORD | -7.62 | 55.1% | **0.56** | 0.77 | 0.76 | 33.2% |
| MOLCHORD-RL$^{dock}$ | **-9.29** | **83.7%** | 0.44 | 0.77 | 0.63 | **59.3%** |
| MOLCHORD-RL | -8.59 | 74.6% | **0.56** | **0.78** | 0.71 | 53.4% |

affinity. Overall, MOLCHORD outperforms all baselines in five key metrics: Vina Dock and High Affinity for binding affinity, QED, SA, and Success Rate for molecular properties, while also maintaining competitive diversity. For binding affinity, the RL-enhanced model achieves the best Vina Dock score and the highest High Affinity, being the first to surpass the 70% threshold and outperforming strong baselines such as FlowSBDD and DecompDiff. Moreover, our gains are substantially larger than those of autoregressive methods, underscoring the importance of the structure encoder in capturing and incorporating structural information.

For molecular properties, both MOLCHORD and MOLCHORD-RL establish state-of-the-art results. On QED, our models perform comparably with strong autoregressive baselines such as Pocket2Mol (Peng et al., 2022) and TamGen (Wu et al., 2024), while achieving the highest SA score (0.78), clearly outperforming diffusion- and flow-based methods. Most importantly, MOLCHORD-RL attains a high Success Rate, reflecting its ability to jointly optimize binding affinity and drug-likeness. These results highlight that our approach effectively leverages the strengths of autoregressive modeling while extending them to drug-like and synthesizable molecule generation. For diversity, MOLCHORD achieves 0.76, second only to the early method GraphBP, which performs poorly on affinity and molecular properties. With RL, diversity decreases slightly—a trade-off also observed in prior works (Cheng et al., 2024)—but remains above 0.70, indicating that our RL improves affinity while still preserving meaningful variation in generation.

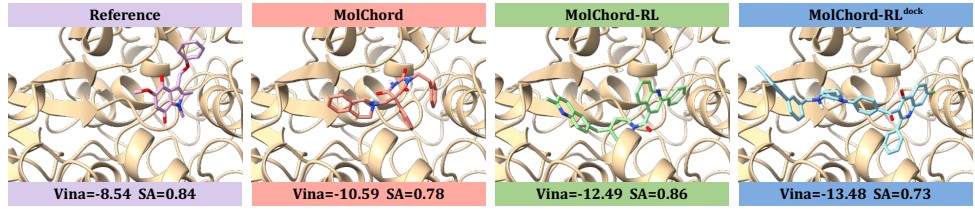

Figure 2: Visualizations of reference molecules and ligands generated by MOLCHORD, MOLCHORD-RL, and MOLCHORD-RL$^{dock}$ for protein pocket 1gg5. Vina score and SA are reported.

Notably, the performance of MOLCHORD-RL$^{dock}$ (Table 1) highlights the trade-off of DPO. While DPO is capable of aggressively improving Vina Dock scores and maintain state-of-the-art SA, it incurs acceptable declines in QED and diversity. Our design instead prioritizes balance, leveraging reward shaping to jointly enforce binding affinity, pharmacological properties, and molecular diversity, achieving strong and stable performance across objectives. Figure 2 provides case studies comparing reference molecules with ligands generated by our approach. We observe that (i) MOLCHORD produces candidates with strong overall quality, (ii) MOLCHORD-RL simultaneously improves binding affinity and molecular properties, and (iii) MOLCHORD-RL$^{dock}$ achieves high

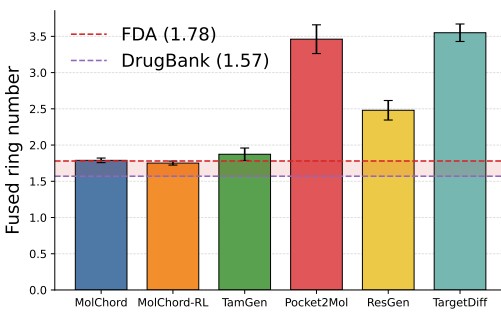

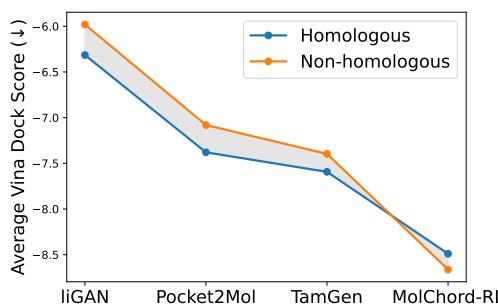

Figure 3: Barplot of the number of fused rings in top-ranked compounds generated by representative methods. For each method, statistics of 1,000 compounds (100 targets × 10 compounds with the highest docking scores) are reported.

Figure 4: OOD generalization: average Vina Dock scores on homologous vs non-homologous proteins for representative methods.

affinity but at the expense of molecular attributes. These examples further illustrate the advantage of balanced optimization in our approach.

**Fused Ring** Fused rings refer to ring systems in which two or more rings share atoms, a structural motif commonly found in bioactive molecules, and often influence both binding and drug-likeness. While fused rings can contribute to favorable binding poses, an excessive number of fused rings is undesirable: prior work such as TamGen (Wu et al., 2024) has shown that an excessive number of fused rings may lead to lower synthetic accessibility (Skoraczyński et al., 2023; Ertl & Schuffenhauer, 2009; Peng et al., 2023), increased cellular toxicity, and decreased developability (Peng et al., 2023; Ritchie & Macdonald, 2009). Indeed, fused rings are known to correlate with QED and SA, making them a useful proxy for chemical plausibility.

Figure 3 shows that our method is the first to match the range of fused ring of approved drugs (see Appendix Table 13 for detailed statistics): MOLCHORD achieves an average of 1.79, close to the FDA reference (1.78), and MOLCHORD-RL further improves to 1.75. For context, DrugBank averages 1.57 fused rings, while representative baselines such as Pocket2Mol, TargetDiff, and ResGen substantially overproduce complex ring systems. These results demonstrate that our approach generates not only high-affinity molecules but also chemically plausible and pharmaceutically relevant candidates, with RL fine-tuning providing additional regularization.

**Out-of-distribution generalization** To further evaluate generalization, we split test proteins into *homologous* and *non-homologous* subsets based on sequence identity with the training set. Pairwise identities were computed using MMseqs2 (Steinegger & Söding, 2017), and proteins sharing more than 30% identity with any training sequence were classified as homologous, yielding 40 homologous and 60 non-homologous cases. Figure 4 reports average Vina Dock scores for both subsets (see Appendix Table 14 for detailed statistics). Prior methods such as liGAN (Ragoza et al., 2022), Pocket2Mol (Peng et al., 2022), and TamGen (Wu et al., 2024) show clear performance drops on non-homologous proteins. In contrast, MOLCHORD-RL not only maintains performance but improves when generalizing to non-homologous proteins ($-8.49 \rightarrow -8.66$, improvement of $+0.17$). We attribute this robustness to the structure encoder, which leverages large-scale pretraining to capture transferable structural features. These results highlight that our approach generalizes beyond training distributions, a critical requirement for real-world drug discovery.

### 4.3 ABLATION STUDY

**Effect of Structure-Sequence Alignment** We further examine the role of alignment design by comparing three variants: (i) **Naïve Alignment**, which directly uses the CrossDocked2020 (Francoeur et al., 2020) training set for alignment; (ii) **Protein–FASTA Alignment**, which performs alignment solely through protein structure–to–FASTA mapping; and (iii) **Full Alignment**, our complete model with molecule–protein–complex alignment. These variants differ only in Stage A,

while Stage B training is kept identical across settings, and all models are evaluated on the Cross-Docked2020 benchmark for comparison. As shown in Table 2, the full alignment achieves the strongest overall performance. For (i) Naïve Alignment, using downstream training dataset directly for Stage A and Stage B leads to overfitting: proteins are not well aligned across structure and sequence space, and the limited chemical exploration results in weaker docking scores and reduced diversity. For (ii) Protein–FASTA Alignment, which aligns proteins at the structure–sequence level and thus alleviates overfitting by better capturing structural–sequential consistency. However, the absence of molecule-related and protein-to-annotation tasks limits chemical space exploration and reduces the benefit of leveraging textual alignment signals. In contrast, (iii) Full Alignment combines protein, molecule, and complex supervision, resulting in the strongest binding affinity and molecular properties. These results highlight the importance of a comprehensive alignment strategy that integrates multiple sources of supervision.

Table 2: The influence of Structure-Sequence Alignment

| Setting | Vina Dock ($\downarrow$) | High Affinity ($\uparrow$) | QED ($\uparrow$) | SA ($\uparrow$) | Diversity ($\uparrow$) | Success Rate ($\uparrow$) |
|---|---|---|---|---|---|---|
| Naïve | -7.38 | 49.8% | 0.55 | 0.77 | 0.74 | 28.6% |
| Protein–FASTA | -7.44 | 50.7% | 0.57 | 0.77 | 0.74 | 31.2% |
| Full | -7.62 | 54.7% | 0.56 | 0.77 | 0.76 | 33.2% |

**Effect of data partitioning** We conduct ablations to disentangle the effect of stratified data usage in SFT and DPO, with results summarized in Table 3. First, let $\mathcal{D}_{\text{full}}$ denote the entire curated CrossDocked2020 dataset. Comparing SFT trained on $\mathcal{D}_{\text{full}}$ versus on the stratified subset $\mathcal{D}_{\text{B}}$, we observe only minor differences: a slight decrease in affinity, accompanied by modest gains in SA and diversity. This indicates that the partitioning procedure itself has limited impact on supervised learning. Second, we investigate the effect of partitioning on preference optimization. Recall that after diversity-based filtering, the dataset used for DPO is denoted as $\mathcal{D}_{\text{DPO}}$. We compare three settings: (i) SFT($\mathcal{D}_{\text{full}}$)+DPO(random), (ii) SFT($\mathcal{D}_{\text{B}}$)+DPO(random), and (iii) SFT($\mathcal{D}_{\text{B}}$)+DPO($\mathcal{D}_{\text{DPO}}$), where "random" denotes a subset drawn uniformly at random from $\mathcal{D}_{\text{DPO}}^{\text{pool}}$ with the same size as $\mathcal{D}_{\text{DPO}}$. This comparison reveals two effects. First, comparing (i) and (ii), we find that separating the preference pool from SFT data yields better DPO performance, with clear gains in affinity. Second, comparing (ii) and (iii), our diversity-based filtering strategy proves effective, resulting in consistent improvements across affinity, molecular properties, and diversity.

Table 3: The influence of data partitioning

| Setting | Vina Dock ($\downarrow$) | High Affinity ($\uparrow$) | QED ($\uparrow$) | SA ($\uparrow$) | Diversity ($\uparrow$) | Success Rate ($\uparrow$) |
|---|---|---|---|---|---|---|
| SFT($\mathcal{D}_{\text{full}}$) | -7.64 | 55.1% | 0.56 | 0.78 | 0.75 | 33.5% |
| SFT($\mathcal{D}_{\text{B}}$) | -7.62 | 54.7% | 0.56 | 0.77 | 0.76 | 33.2% |
| SFT($\mathcal{D}_{\text{full}}$)+DPO(random) | -8.22 | 67.5% | 0.54 | 0.77 | 0.68 | 42.1% |
| SFT($\mathcal{D}_{\text{B}}$)+DPO(random) | -8.44 | 71.6% | 0.53 | 0.77 | 0.68 | 47.1% |
| SFT($\mathcal{D}_{\text{B}}$)+DPO($\mathcal{D}_{\text{DPO}}$) | -8.59 | 74.6% | 0.56 | 0.78 | 0.71 | 53.4% |

## 4.4 ADMET-AWARE REWARD INTEGRATION

To further assess the extensibility of our optimization framework, we investigate the effect of incorporating ADMET-related constraints by conducting an additional experiment on blood–brain barrier (BBB) penetration, which reflects the distribution aspect (the "D" in ADMET) of molecules. The BBB labels come from the BBB_Martins dataset (Martins et al., 2012). BBB permeability is predicted using ADMET-AI (Swanson et al., 2024) as a binary classifier, where molecules with predicted values above 0.5 are considered permeable. We evaluate both MOLCHORD and MOL-CHORD-RL on 10,000 molecules generated from the CrossDocked2020 test pockets and report the corresponding BBBP statistics.

We further extend the reinforcement-learning objective by combining docking affinity with the predicted BBBP signal. The modified reward is:

$$R(M, P^{\text{pock}}) = -\left(S_{\text{Vina}}(M, P^{\text{pock}}) - \lambda_{\text{bbbp}} \cdot \text{BBBP}\right),$$

Table 4: Comparison of ADMET-aware optimization on BBB permeability. The first two rows correspond to MOLCHORD (no RL) and MOLCHORD-RL (Affinity + Fused Rings), while the third row reports the variant trained with the combined Affinity + BBBP reward.

| Reward Components | Vina Dock ($\downarrow$) | BBBP | QED ($\uparrow$) | SA ($\uparrow$) | Lipinski ($\uparrow$) | Diversity ($\uparrow$) | Success Rate ($\uparrow$) |
|---|---|---|---|---|---|---|---|
| None | -7.62 | 0.688 | 0.56 | 0.77 | 4.66 | 0.76 | 33.2% |
| Affinity+Fused Rings | -8.59 | 0.683 | 0.56 | 0.78 | 4.72 | 0.71 | 53.4% |
| Affinity+BBBP | -8.54 | 0.781 | 0.54 | 0.77 | 4.59 | 0.68 | 49.7% |

where BBBP $\in \{0, 1\}$ denotes the predicted permeability and $\lambda_{\text{bbbp}}$ is set to 2. We retrain the RL stage with this combined reward and evaluate the generated molecules using the same protocol.

The comparative results for MOLCHORD, MOLCHORD-RL, and the affinity+BBBP variant are summarized in Table 4. The combined reward leads to a moderate decrease in docking affinity but substantially improves BBB penetration, demonstrating that ADMET-aware signals can be readily incorporated into our framework and may provide a viable path toward multi-objective drug-design optimization.

## 5 CONCLUSION

In this paper, we introduced MOLCHORD, a framework for SBDD that combines a diffusion-based structure encoder with an autoregressive generator. The framework enhances alignment by linking proteins with FASTA and descriptions, molecules with SMILES and descriptions, and complexes with paired FASTA–SMILES representations. To enable effective preference optimization, we proposed a stratified data split and constructed a curated DPO dataset, which proved critical for improving model performance. Beyond binding affinity, our method effectively balances diversity and pharmacological properties, both of which are crucial for drug discovery. The results highlight the potential of our approach as a general framework for SBDD.

A current limitation is that our work assumes a rigid protein structure during ligand generation, preventing it from modeling induced fit or conformational flexibility. Incorporating protein dynamics and extending the structure encoder toward 3D generation represent promising directions for future work. Overall, MOLCHORD offers a solid foundation that can be further expanded toward more flexible and unified sequence-structure ligand modeling.

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

# A  ARCHITECTURE

We provide detailed architectural descriptions for each component of MOLCHORD—the structure encoder, sequence generator, adapter, and VAE—with hyperparameters presented separately for each module.

## A.1  STRUCTURE ENCODER

Our structure encoder consists of two components: a sequence module and a structure module. The sequence module(encoder) maps proteins, molecules, and complexes into feature representations that support both intra-modal and cross-modal interactions. Meanwhile, the diffusion-based structure module (decoder) captures residue and atom distributions, yielding 3D coordinates and enriching representations with structural context. Detailed architectural hyperparameters are provided in Table 5.

Table 5: Hyperparameters of Structure Encoder.

| Hyperparameters | Sequence Module | Structure Module |
|---|---|---|
| Number of layers | 32 | 16 |
| Hidden size | 2048 | 2048 |
| FFN dimension | 8192 | 8192 |
| Attention heads | 32 | 32 |

**Sequence Module.**  The sequence module separately processes protein sequences at the residue level and molecular graphs at the atom level, representing each as tokens. A standard Transformer encoder is applied, where molecule tokens are augmented with learnable attention biases derived from their 2D topology, enabling the model to capture chemical connectivity. The resulting embeddings capture intra-protein, intra-molecule, and protein–molecule interactions, providing a comprehensive feature representation for subsequent modeling.

**Structure Module.**  The structure module is implemented as a Diffusion Transformer (DiT) (Peebles & Xie, 2023) that denoises the 3D coordinates of protein residues and molecular atoms. Coordinates are denoised under the conditioning of sequence-module representations, after being projected from 3D into a higher-dimensional latent space. Notably, ligand atoms require additional attention biases derived from bond connectivity. Through this design, this module refines noisy coordinates into chemically consistent structures, yielding enriched representations that couple spatial detail with sequence context for subsequent modeling.

## A.2  SEQUENCE GENERATOR

We implement a reproduction of NatureLM-1B (Xia et al., 2025). The tokenizer is initialized from the LLaMA-3 vocabulary (Dubey et al., 2024) (128,256 general-purpose tokens) and extended with a minimal set of domain-specific tokens: 26 for protein FASTA sequences, 1,401 for molecular SMILES strings, and four special markers "⟨mol⟩", "⟨/mol⟩", "⟨protein⟩", and "⟨/protein⟩" to indicate modality boundaries. Architectural hyperparameters are given in Table 3. The model is trained with a next-token prediction objective on both single-domain corpora (text, proteins, molecules) and cross-modal corpora (protein–text, molecule–text, protein–molecule–text), enabling it to retain general language modeling capacity while incorporating biomolecular semantics. The corresponding architectural hyperparameters are listed in Table 6.

## A.3  ADAPTER AND VAE

**Adapter**  The adapter module provides a lightweight interface for injecting structural features into the language model. It adopts a gated MLP: input representations are processed by a gating projection and an up-projection, with the gated branch passing through a non-linear activation and combined element-wise with the up-projected features. A down-projection then maps the fused

Table 6: Hyperparameters of Sequence Generator.

| Hyperparameters | Value |
|---|---|
| Vocabulary size | 129,687 |
| Number of layers | 16 |
| Hidden size | 2048 |
| FFN dimension | 5504 |
| Attention heads | 32 |

representation back to the hidden space, enabling efficient alignment with minimal additional parameters. Table 7 reports the detailed architectural hyperparameters.

Table 7: Hyperparameters of Adapter.

| Hyperparameters | Value |
|---|---|
| Input dimension | 2048 |
| Intermediate dimension | 2048 |
| Output dimension | 2048 |

**VAE**   The variational encoder maps complex representations into a latent Gaussian space using two MLPs that predict the mean and log-variance of the posterior. During training, it is only activated in Stage B and Stage C. The latent distribution of complex from structure encoder are injected as noise into the feature of the corresponding protein from structure encoder, thereby perturbing protein features and improving robustness. Architectural hyperparameters are summarized in Table8.

Table 8: Hyperparameters of VAE.

| Hyperparameters | Value |
|---|---|
| Input dimension | 2048 |
| Latent dimension | 2048 |

## B   IMPLEMENTATION DETAILS

### B.1   STRUCTURE ENCODER PRE-TRAINING

The architecture of the structure encoder follows the Elucidated Diffusion Model (EDM) (Karras et al., 2022), a Transformer variant that integrates geometric information and has also been adopted in AlphaFold3 (Abramson et al., 2024).

**Dataset Construction**   The structure encoder was pre-trained on 78M protein structures derived from both experimentally solved PDB entries (Berman et al., 2000) and predicted structures from AlphaFoldDBB (Varadi et al., 2024). The exact filtering protocol is as follows:

For PDB, we use the PDB20210930 snapshot and adopt the same quality-control criteria as AlphaFold3 (Abramson et al., 2024): (a) Structures containing >300 chains are removed; (b) Structures with resolution worse than 9Å are discarded; (c) Entries with fewer than 4 amino acids are excluded.

For AlphaFoldDB, which contains over 200M predicted protein models, we apply two layers of filtering to reduce redundancy and ensure structural reliability: (a) 90% sequence-identity clustering (MMseqs2) and retain only cluster representatives; (b) A minimum global pLDDT threshold of 70 to remove low-confidence predictions. After applying these filters, we obtain approximately 78 million non-redundant, quality-controlled protein structures for encoder pre-training. This ensures that low-quality or noisy structural predictions do not introduce bias into the learned representations.

**Full Configuration**   To facilitate reproducibility, we provide the complete set of hyperparameters used to train the 3B-parameter encoder, as summarized in Table 9. All pre-training follows standard large-scale protein-modeling practices, and we do not rely on proprietary optimization tricks.

Table 9: Hyperparameters used for pre-training the structure encoder.

| Component | Configuration |
|---|---|
| Numerical precision | bfloat16 (bf16) |
| Global batch size | 4096 |
| Optimizer | AdamW |
| Peak learning rate | $1 \times 10^{-4}$ |
| Learning rate schedule | Cosine decay |
| Warm-up steps | 2000 |
| Total training steps | 200k |
| Compute | 128×NVIDIA A100 (80GB) |
| Training duration | ~14 days |

### B.2   SEQUENCE GENERATOR PRE-TRAINING

Following NatureLM (Xia et al., 2025), we pre-train the sequence generator in three stages on 64 NVIDIA A100 GPUs over 14 days.

(i) Stage 1: training from scratch on 300B SlimPajama tokens with the original LLaMA-3 vocabulary (128,256 tokens), using AdamW with learning rate $3 \times 10^{-4}$, batch size 4,096, context length 8,192, cosine decay, for 18K steps.

(ii) Stage 2: extending the tokenizer with domain-specific scientific tokens (SMILES, FASTA, special modality markers) and training for 4K steps while updating only the new embeddings.

(iii) Stage 3: full-model continued pretraining on 80B tokens from mixed-domain corpora, including both interleaved cross-modal data (text–molecule, text–protein, protein–molecule) and single-domain corpora. The training data sources cover C4 (Raffel et al., 2020), PubChem (Kim et al., 2023), UniRef90 (Suzek et al., 2007), Swiss-Prot (Boutet et al., 2007), ZINC (Irwin & Shoichet, 2005), among others. A reduced learning rate of $1 \times 10^{-4}$ is used for 15K steps.

### B.3   POST-TRAINING PROCEDURES

**Alignment**   Our alignment is implemented on a large-scale dataset of 1.1M instances (676K for proteins, 316K for molecules and 94K for complexes). The model is optimized with a learning rate of $1 \times 10^{-4}$, a batch size of 512, and 60K training steps, while keeping the backbone frozen and updating only the adapter parameters. Training was conducted on 32 A100 GPUs for 5 days.

**Supervised Fine-tuning**   For supervised fine-tuning, we use 100K examples from the Cross-Docked2020 dataset. The model is optimized with a learning rate of $1 \times 10^{-5}$, a batch size of 128, and 15K training steps. The KL loss coefficient $\beta_{\text{vae}}$ is set to 0.1, and the VAE latent size is 2048. Training was performed on 8 A100 GPUs for approximately 30 hours.

**Reinforcement Learning**   For reinforcement learning with Direct Preference Optimization (DPO), we train on the $\mathcal{D}_{\text{DPO}}$ set consisting of 979 examples. The model is optimized with a learning rate of $5 \times 10^{-7}$ and a batch size of 8 for a single epoch, such that each sample is seen only once. The KL penalty coefficient $\beta_{\text{vae}}$ is set to 0.1 and the VAE latent size to 2048, identical to the SFT setting. Training is highly efficient and completes within 4 hours on 8 A100 GPUs with 112 vCPUs.

For online DPO, each protein pocket is used to generate 32 candidate molecules. Among valid generations, 5 are selected for docking, and the rewards described in the main text are used to construct best–worst preference pairs. The DPO loss employs $\beta_{\text{dpo}} = 0.1$ to scale the advantage term in the preference objective. Sampling is performed with temperature 1.5 and top-$p = 0.95$ to encourage diversity, while a fused-ring penalty with weight $\lambda = 0.5$ is applied to regularize chemical plausibility.

**Inference**   During inference, we sample molecules with temperature set to 1.5, a maximum generation length of 256 tokens, and top-$p = 0.95$. For each protein pocket, at least 100 valid candidate molecules are generated to ensure sufficient diversity for downstream evaluation.

## C   EXPERIMENT DETAILS AND SUPPLEMENTARY RESULTS

### C.1   EXPERIMENT DETAILS

**Distribution Analysis of CrossDocked2020**   We visualize the distribution of candidate ligands per target in the CrossDocked2020 dataset, as shown in Figure 5.

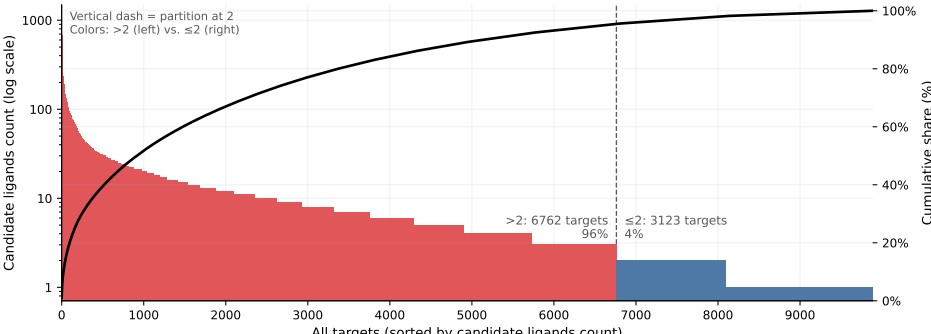

Figure 5: Distribution of candidate ligands per target in the CrossDocked2020 dataset. Targets are sorted by ligand count, with a dashed line marking the partition at 2 ligands, where the red and blue regions correspond to $\mathcal{D}_B$ and $\mathcal{D}_C$, respectively.

**Training Dynamics of the RL Stage**   We track the optimization trajectory of the RL stage over 5 epochs (123 steps per epoch). The training loss curve, shown in Figure 6, indicates stable convergence throughout the RL process. Quantitative metrics at representative checkpoints (0.5, 1, 2, and 3 epochs) are summarized in Table 10, and the evolution of QED, SA, Diversity, and Vina Dock is plotted in Figure 7. The curves suggest a gradual shift in optimization behavior: early epochs improve affinity and success rate while keeping QED, SA, and Lipinski scores stable, whereas later epochs place more emphasis on affinity, with modest changes observed in diversity and drug-likeness.

Table 10: Metrics evaluated at different RL training epochs.

| Epoch | Vina Dock ($\downarrow$) | High Affinity ($\uparrow$) | QED ($\uparrow$) | SA ($\uparrow$) | Lipinski ($\uparrow$) | Diversity ($\uparrow$) | Success Rate ($\uparrow$) |
|---|---|---|---|---|---|---|---|
| 0.5 | -8.01 | 62.8% | 0.55 | 0.77 | 4.62 | 0.73 | 39.3% |
| 1 (MolChord-RL) | -8.59 | 74.6% | 0.56 | 0.78 | 4.72 | 0.71 | 53.4% |
| 2 | -9.18 | 81.7% | 0.42 | 0.77 | 4.37 | 0.63 | 52.2% |
| 3 | -9.56 | 86.7% | 0.38 | 0.77 | 4.31 | 0.61 | 53.1% |

**SA score**   Note that the SA score is originally defined on a scale from 1 to 10 (Ertl & Schuffenhauer, 2009), with lower values indicating greater synthesizability. Consistent with prior work on pocket-aware 3D drug design (Guan et al., 2023a), we apply a linear transformation, $SA = (10 - SA_{\text{origin}})/9 \in [0, 1]$, so that higher values correspond to better synthesizability.

**Generation Setup**   In the structure-based drug design experiments, each baseline method generates no more than 100 molecules for a given protein pocket. In comparison, MOLCHORD produces exactly 100 unique molecules per pocket, enforcing a stricter evaluation protocol.

**Docking Details**   To evaluate docking, we convert generated SMILES strings into 3D molecular conformations. Molecules are first parsed with OpenBabel (O'Boyle et al., 2011) to obtain an initial structure, which is then processed with RDKit for conformer generation. We apply distance-geometry embedding with a fixed random seed, followed by MMFF optimization. If embedding

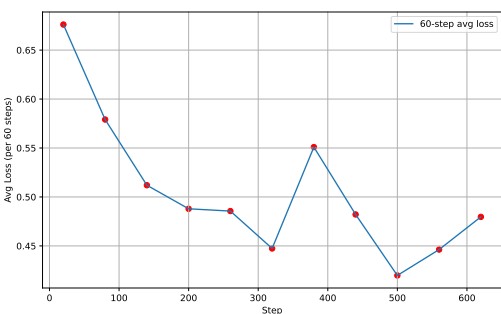 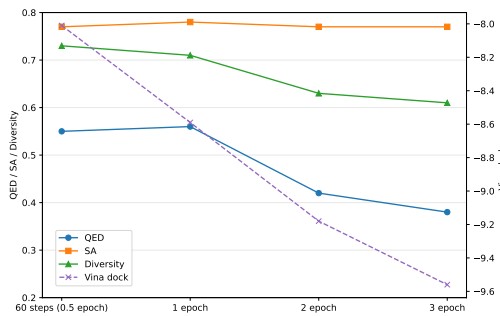

Figure 6: Training loss curve of the RL stage, with one point recorded every 60 training steps. The curve shows stable convergence across all 5 epochs.

Figure 7: Evolution of QED, SA, Diversity, and Vina Dock across representative RL epochs. The curves illustrate the trade-offs between affinity, drug-likeness, and diversity during RL optimization.

fails, a 2D coordinate initialization is used as fallback. For docking, since our model does not generate binding poses directly, we use the center of the reference ligand as the pocket center. Docking is then performed with AutoDock Vina.

## C.2    ADDITIONAL RESULTS

In this subsection, we provide supplementary experimental results that could not be included in the main text. These include extended tables and figures for the main results, additional ablation studies, and further case study analyses.

### C.2.1    MAIN RESULTS

**Complete Results**    We further include three representative structure-based generation methods—DrugGPS (Zhang & Liu, 2023), FLAG (Zhang et al., 2023b), and Frag2Seq (Fu et al., 2024)—in our main comparison. DrugGPS generates 3D ligands by predicting sub-pocket prototypes and assembling atoms around these learned structural motifs. FLAG constructs molecules in 3D by sequentially placing fragments according to fragment-level priors and geometric constraints. Frag2Seq adopts a geometry- and fragment-aware tokenization scheme and autoregressively decodes fragment sequences into complete molecular structures. These works predominantly report *Vina Score* computed on raw generated conformers, whereas our evaluation relies on *Vina Dock*, which measures docking affinity after re-docking the generated molecules against the target pocket. Since these two metrics are not directly comparable, we re-evaluate DrugGPS and FLAG using their publicly released inference outputs under the same Vina Dock settings as MOLCHORD. Lipinski is also reported for completeness, it is computed following the implementation in Pocket2Mol (Peng et al., 2022). The consolidated results are presented in Table 1.

**SBDD Benchmark Results**    To provide an additional assessment under a widely used structure-based benchmark, we follow the protocol of the Zheng et al. (2024) and evaluate MOLCHORD-RL on the same seven targets (1IEP, 3EML, 3NY8, 4RLU, 4UNN, 5MO4, 7L11). For each target, we generate 1000 molecules using the same generation settings as the benchmark and report the top-10 docking scores averaged over each target. As shown in Table 12, MOLCHORD-RL achieves state-of-the-art performance on 3NY8, 4RLU, and 5MO4, ranks second on the remaining four targets, and obtains the best average score across all seven targets, illustrating its strong overall performance on this benchmark.

**Median Vina Energy**    Figure 8 shows the median Vina energy of the proposed model, compared with TargetDiff, Pocket2Mol and TamGen, three representative methods in target-aware molecule generation. We observe that MOLCHORD surpasses these baseline models and generates molecules with the highest binding affinity for 50% of the protein targets in the test set.

Table 11: Main results comparing MOLCHORD with representative pocket-aware drug design baselines. For methods lacking publicly available outputs, only the reported metrics are included, and unavailable values are left blank. Top-2 results are marked in **bold** and underline, respectively.

| Methods | Vina Dock ($\downarrow$) | High Affinity ($\uparrow$) | QED ($\uparrow$) | SA ($\uparrow$) | Lipinski ($\uparrow$) | Diversity ($\uparrow$) | Success Rate ($\uparrow$) |
|---|---|---|---|---|---|---|---|
| Reference | -7.45 | - | 0.48 | 0.73 | 4.34 | - | 25.0% |
| LiGAN | -6.33 | 21.1% | 0.39 | 0.59 | - | 0.66 | 3.9% |
| GraphBP | -4.80 | 14.2% | 0.43 | 0.49 | 4.88 | **0.79** | 0.1% |
| AR | -6.75 | 37.9% | 0.51 | 0.63 | - | 0.70 | 7.1% |
| Pocket2Mol | -7.15 | 48.4% | 0.56 | 0.74 | 4.94 | 0.69 | 24.4% |
| TamGen | -7.48 | 52.6% | 0.56 | 0.77 | 4.88 | 0.75 | 32.4% |
| TargetDiff | -7.80 | 58.1% | 0.48 | 0.58 | 4.59 | 0.72 | 10.5% |
| DecompDiff | -8.39 | 64.4% | 0.45 | 0.61 | 4.49 | 0.68 | 24.5% |
| MolCRAFT | -7.92 | 59.1% | 0.50 | 0.69 | - | 0.72 | 26.8% |
| FlowSBDD | -8.50 | 63.4% | 0.47 | 0.51 | - | 0.75 | - |
| FLAG | -7.06 | 47.8% | 0.49 | 0.70 | 4.66 | 0.70 | 16.9% |
| DrugGPS | -7.48 | 42.1% | 0.47 | 0.63 | 4.50 | 0.74 | 14.1% |
| Frag2Seq | - | 65.3% | **0.65** | 0.64 | **4.99** | 0.71 | - |
| MOLCHORD | -7.62 | 55.1% | 0.56 | 0.77 | 4.66 | 0.76 | 33.2% |
| MOLCHORD-RL$^{dock}$ | **-9.29** | **83.7%** | 0.44 | 0.77 | 4.48 | 0.63 | **59.3%** |
| MOLCHORD-RL | -8.59 | 74.6% | 0.56 | **0.78** | 4.72 | 0.71 | 53.4% |

Table 12: Top-10 averaged docking scores on the seven SBDD benchmark targets. Top-2 results are marked in **bold** and underline, respectively.

| Model | 1IEP | 3EML | 3NY8 | 4RLU | 4UNN | 5MO4 | 7L11 | Avg |
|---|---|---|---|---|---|---|---|---|
| 3DSBDD(Luo et al., 2021) | -9.05±0.38 | -10.02±0.15 | -10.10±0.24 | -9.80±0.55 | -8.23±0.30 | -8.71±0.45 | -8.47±0.18 | -9.20 |
| AutoGrow4(Spiegel & Durrant, 2020) | **-13.23±0.11** | **-13.03±0.09** | -11.70±0.00 | -11.20±0.00 | -11.14±0.12 | -10.38±0.27 | -8.84±0.33 | -11.36 |
| Pocket2Mol(Peng et al., 2022) | -10.17±0.53 | -12.25±0.27 | -11.89±0.16 | -10.57±0.12 | **-12.20±0.34** | -10.07±0.62 | **-9.74±0.38** | -10.98 |
| PocketFlow(Jiang et al., 2024) | -12.49±0.70 | -9.25±0.29 | -8.56±0.35 | -9.65±0.25 | -7.90±0.78 | -7.80±0.42 | -8.35±0.31 | -9.14 |
| ResGen(Zhang et al., 2023a) | -10.97±0.29 | -9.25±0.95 | -10.96±0.42 | -11.75±0.42 | -9.41±0.23 | -10.34±0.39 | -8.74±0.24 | -10.20 |
| DST(Fu et al., 2021a) | -10.95±0.57 | -10.67±0.24 | -10.54±0.22 | -10.88±0.37 | -9.71±0.19 | -10.03±0.36 | -8.33±0.41 | -10.16 |
| GraphGA(Jiang et al., 2024) | -10.03±0.41 | -9.89±0.25 | -9.94±0.15 | -10.22±0.39 | -9.32±0.51 | -9.29±0.20 | -7.75±0.32 | -9.49 |
| MIMOSA(Fu et al., 2021b) | -10.96±0.57 | -10.69±0.24 | -10.51±0.23 | -10.81±0.39 | -9.66±0.25 | -10.02±0.36 | -8.33±0.41 | -10.14 |
| MolDQN(Zhou et al., 2019) | -6.73±0.12 | -6.51±0.15 | -7.09±0.16 | -6.79±0.26 | -5.92±0.26 | -6.27±0.10 | -6.87±0.20 | -6.60 |
| Pasithea(Shen et al., 2021) | -10.86±0.29 | -10.31±0.09 | -10.69±0.27 | -10.92±0.35 | -9.69±0.32 | -9.77±0.21 | -8.06±0.22 | -10.04 |
| REINVENT(Olivecrona et al., 2017) | -9.87±0.31 | -9.48±0.39 | -9.61±0.36 | -9.69±0.29 | -8.70±0.25 | -8.92±0.38 | -7.25±0.21 | -9.07 |
| SCREENING(Zheng et al., 2024) | -10.86±0.26 | -10.90±0.54 | -10.73±0.45 | -10.86±0.22 | -9.80±0.23 | -9.91±0.30 | -8.15±0.26 | -10.17 |
| SELFIES-VAE-BO(Gómez-Bombarelli et al., 2018) | -10.15±0.60 | -9.76±0.12 | -9.99±0.28 | -9.99±0.23 | -10.00±0.23 | -9.18±0.39 | -7.75±0.22 | -9.41 |
| SMILES GA(Yoshikawa et al., 2018) | -9.56±0.17 | -9.56±0.37 | -10.00±0.26 | -9.61±0.19 | -8.80±0.20 | -9.21±0.23 | -7.54±0.32 | -9.18 |
| SMILES LSTM HC(Brown et al., 2019) | -10.38±0.21 | -10.30±0.15 | -10.19±0.12 | -10.49±0.49 | -9.36±0.17 | -9.71±0.43 | -7.90±0.26 | -9.76 |
| SMILES-VAE-BO(Gómez-Bombarelli et al., 2018) | -9.93±0.22 | -9.78±0.10 | -9.96±0.29 | -10.05±0.20 | -9.03±0.30 | -9.18±0.39 | -7.74±0.25 | -9.38 |
| MOLCHORD-RL (Ours) | -12.69±0.24 | -12.39±0.53 | **-13.02±0.23** | **-12.96±0.33** | -11.38±0.21 | **-11.34±0.18** | -9.50±0.16 | **-11.90** |

**Fused Ring** The quantitative results of the fused-ring analysis is reported in Table 13.

Table 13: Fused ring statistics for different generation methods.

| Method | Fused ring count |
|---|---|
| TamGen | 1.87 |
| Pocket2Mol | 3.46 |
| ResGen | 2.48 |
| TargetDiff | 3.55 |
| MOLCHORD | 1.79 |
| MOLCHORD-RL | 1.75 |

**OOD Results** The quantitative results of the OOD evaluation are reported in Table 14. The full list of PDB IDs, comprising 40 homologous and 60 non-homologous cases, is provided below:

Homologous (40 cases): 4aaw, 4yhj, 14gs, 1fmc, 3g51, 2jjg, 4g3d, 5bur, 5q0k, 2azy, 5i0b, 1phk, 1djy, 5ll1v, 4zfa, 4f1m, 4iwq, 5ngz, 1d7j, 4u5s, 3pdh, 1umd, 4pxz, 2gns, 1ai4, 5mma, 2cy0, 5d7n, 5mgl, 5aeh, 4xli, 3o96, 3hy9, 4bel, 4aua, 2f2c, 3chc, 1k9t, 1jn2, 4azf.

Non-homologous (60 cases): 2z3h, 2v3r, 4rn0, 3daf, 1a2g, 5w2g, 3dzh, 1coy, 2rhy, 2pqw, 3gs6, 1r1h, 1dxo, 1gg5, 5b08, 4keu, 4q8b, 2rma, 3b6h, 2zen, 4p6p, 3u5y, 4tqr, 4lfu, 3jyh, 1l3l, 1e8h, 2e24, 2hcj, 3kc1, 4ja8, 4iiy, 3v4t, 3tym, 4d7o, 3ej8, 1rs9, 4kcq, 3w83, 2e6d, 4rv4, 1h36, 4gvd, 4tos, 4h3c, 4rlu, 3l3n, 5tjn, 5liu, 4qlk, 3nfb, 4m7t, 3u9f, 1h0i, 4z2g, 3af2, 3li4, 3pnm, 1afs, 2pc8.

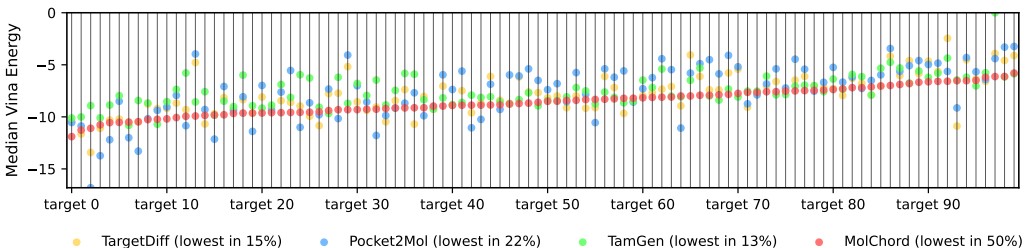

Figure 8: Median Vina energy for different generated molecules (TargetDiff, Pocket2Mol, TamGen, MolChord) across 100 testing samples, sorted by the median Vina energy of molecules generated from MolChord.

Table 14: Comparison of average scores on homologous vs. non-homologous pockets, with $\Delta$ denoting their difference.

| Method | Homologous (avg) | Non-homologous (avg) | $\Delta$ |
|---|---|---|---|
| liGAN | -6.31 | -5.98 | -0.33 |
| Pocket2Mol | -7.38 | -7.08 | -0.30 |
| TamGen | -7.59 | -7.40 | -0.20 |
| MolChord-RL | -8.49 | -8.66 | **+0.17** |

**Efficiency**   MolChord also demonstrates superior generation efficiency. Following the evaluation protocol of Frag2Seq (Fu et al., 2024), we measure the time required to generate 100 molecules per target on a single A100 GPU. As summarized in Table 15, prior structure-based generation methods typically require from tens of seconds to several minutes for this workload, with most methods exceeding 1000 seconds. In contrast, MolChord produces 100 compounds in approximately 5 seconds by leveraging highly parallelized batched autoregressive decoding (batch size = 128). This substantial speed advantage underscores the practicality of MolChord for large-scale or high-throughput molecular generation.

Table 15: Comparison of generation time for producing 100 molecules per target. All methods are evaluated on a single GPU following the protocol of Frag2Seq. Lower is better ($\downarrow$).

| Method | Time (s, $\downarrow$) |
|---|---|
| 3D-SBDD | 15986.4 |
| Pocket2Mol | 2827.3 |
| GraphBP | 1162.8 |
| TargetDiff | 3428.0 |
| DecompDiff | 6189.0 |
| DiffSBDD | 629.9 |
| FLAG | 1289.1 |
| DrugGPS | 1007.8 |
| Lingo3DMol | 1481.9 |
| Frag2Seq | 48.8 |
| MolChord | 5.0 |

### C.2.2   ABLATION STUDIES

**Effect of VAE**   The effect of incorporating the VAE is shown in Table 16. We observe consistent gains across all evaluation metrics when the VAE is included, with particularly notable improvements in affinity-related measures. This can be attributed to the stochasticity introduced by the latent variables, which encourages broader exploration of the chemical space and enhances both molecular diversity and model robustness.

Table 16: The influence of VAE

| Setting | Vina Dock ($\downarrow$) | High Affinity ($\uparrow$) | QED ($\uparrow$) | SA ($\uparrow$) | Diversity ($\uparrow$) | Success Rate ($\uparrow$) |
|---|---|---|---|---|---|---|
| MOLCHORD w/o VAE | -7.44 | 50.2% | 0.55 | 0.76 | 0.75 | 29.5% |
| MOLCHORD | -7.62 | 54.7% | 0.56 | 0.77 | 0.76 | 33.2% |

**Effect of Global Protein Structure** To assess the contribution of global protein information, we perform an ablation in which the structure encoder receives only the pocket atoms while keeping all other components and training settings unchanged. As shown in Table 17, using only the pocket atoms yields similar molecular property metrics (QED, SA, Lipinski) and diversity, but results in lower docking performance and success rate compared with using the full protein. This indicates that global structural context provides complementary binding-relevant cues that are not fully captured by the pocket alone.

Table 17: Ablation on the effect of global protein structure. The structure encoder is given either the full protein or only the pocket atoms.

| Structure Encoder Input | Vina Dock ($\downarrow$) | High Affinity ($\uparrow$) | QED ($\uparrow$) | SA ($\uparrow$) | Lipinski ($\uparrow$) | Diversity ($\uparrow$) | Success Rate ($\uparrow$) |
|---|---|---|---|---|---|---|---|
| Pocket Only | -7.22 | 45.6% | 0.56 | 0.77 | 4.66 | 0.77 | 25.7% |
| Full Protein (MOLCHORD) | -7.62 | 55.1% | 0.56 | 0.77 | 4.66 | 0.76 | 33.2% |

**Effect of Reward Components.** We evaluate several alternative RL reward designs by replacing the fused-ring term with QED, SA, or multi-property objectives, as well as a success-rate reward following (Hu et al., 2025). All variants share the same affinity component and are trained under identical settings. The reward formulations are:

$$\textbf{Affinity + QED: } R = -\left(S_{\text{Vina}} - \lambda_{\text{qed}} \cdot \text{QED}\right)$$

$$\textbf{Affinity + SA: } R = -\left(S_{\text{Vina}} - \lambda_{\text{sa}} \cdot \text{SA}\right)$$

$$\textbf{Affinity + QED + SA: } R = -\left(S_{\text{Vina}} - \lambda_{\text{qed}} \cdot \text{QED} - \lambda_{\text{sa}} \cdot \text{SA}\right)$$

$$\textbf{Affinity + QED + SA + FR: } R = -\left(S_{\text{Vina}} - \lambda_{\text{qed}} \cdot \text{QED} - \lambda_{\text{sa}} \cdot \text{SA} + \lambda_{\text{fr}} \max(0, \#\texttt{fused\_ring}(M) - 2)\right)$$

We use $\lambda_{\text{qed}} = 5$, $\lambda_{\text{sa}} = 5$, and $\lambda_{\text{fr}} = 0.5$ for all experiments. As shown in Table 18, QED- and SA-based rewards improve their targeted properties but often reduce docking affinity and increase fused-ring counts. Both the combined Affinity+QED+SA+FR reward and Affinity+FR design (MOLCHORD) provide strong overall performance, representing two effective optimization choices depending on the desired trade-off.

Table 18: Ablation on the effect of different reward components used during RL optimization.

| Reward Components | Vina Dock ($\downarrow$) | High Affinity ($\uparrow$) | QED ($\uparrow$) | SA ($\uparrow$) | Lipinski ($\uparrow$) | Diversity ($\uparrow$) | Success Rate ($\uparrow$) | Top10 Fused Rings |
|---|---|---|---|---|---|---|---|---|
| Affinity + QED | -8.45 | 72.3% | 0.57 | 0.78 | 4.68 | 0.70 | 49.2% | 1.90 |
| Affinity + SA | -8.39 | 70.0% | 0.54 | 0.79 | 4.63 | 0.70 | 47.3% | 1.82 |
| Affinity + QED + SA | -8.45 | 72.0% | 0.57 | 0.80 | 4.73 | 0.71 | 50.3% | 1.87 |
| Affinity + QED + SA + FR | -8.56 | 74.3% | 0.58 | 0.80 | 4.73 | 0.71 | 53.2% | 1.86 |
| Follow 3DMolFormer | -8.20 | 66.5% | 0.56 | 0.77 | 4.66 | 0.70 | 43.8% | 1.87 |
| MOLCHORD-RL (Affinity + FR) | -8.59 | 74.6% | 0.56 | 0.78 | 4.72 | 0.71 | 53.4% | 1.75 |

**Effect of Fused-ring Penalty** To examine the effect of the fused-ring penalty coefficient $\lambda$, we evaluate $\lambda \in \{0, 0.2, 0.5, 0.8, 1.0\}$ under the same DPO curation pipeline and with identical training steps, as summarized in Table 19. For each setting, we report docking performance, molecular properties, diversity, and fused-ring statistics. Lipinski-related metrics are computed following the implementation in Pocket2Mol (Peng et al., 2022).

Across this range, the model exhibits broadly stable behavior: smaller coefficients (e.g., 0 and 0.2) provide strong docking performance but allow a higher frequency of fused-ring structures, while larger coefficients (e.g., 0.8 and 1.0) more effectively suppress fused rings with only moderate changes in docking and molecular property metrics. Overall, the system does not appear highly

Table 19: Effect of fused-ring penalty coefficient $\lambda$

| Penalty $\lambda$ | Vina Dock ($\downarrow$) | High Affinity ($\uparrow$) | QED ($\uparrow$) | SA ($\uparrow$) | Lipinski ($\uparrow$) | Diversity ($\uparrow$) | Success Rate ($\uparrow$) | Top10 Fused Rings |
|---|---|---|---|---|---|---|---|---|
| 0 | -8.72 | 77.7% | 0.53 | 0.77 | 4.57 | 0.67 | 52.7% | 1.99 |
| 0.2 | -8.74 | 77.4% | 0.52 | 0.78 | 4.57 | 0.67 | 53.2% | 1.92 |
| 0.5 | -8.59 | 74.6% | 0.56 | 0.78 | 4.72 | 0.71 | 53.4% | 1.75 |
| 0.8 | -8.47 | 71.9% | 0.55 | 0.78 | 4.62 | 0.71 | 48.9% | 1.61 |
| 1.0 | -8.37 | 69.6% | 0.53 | 0.77 | 4.57 | 0.70 | 46.2% | 1.71 |

sensitive within this interval, and multiple values (such as 0.2 or 0.8) achieve reasonable trade-offs. We adopt $\lambda = 0.5$ as a middle-ground choice that maintains balanced performance across docking scores, fused-ring control, molecular properties, and diversity.

## D  PROMPT

All pre-training and fine-tuning tasks are formulated as text-augmented generation: structured entities (proteins, molecules, or complexes) are encoded into feature vectors by the structure encoder and injected into reserved slots of the language model's input embeddings. Placeholders marked as "$\langle$3d protein$\rangle$", "$\langle$3d molecule$\rangle$", or "$\langle$3d complex$\rangle$" are routed to the structure encoder rather than tokenized, and their features replace the corresponding placeholder tokens in the prompt embedding.

### D.1  PROTEIN ALIGNMENT PROMPTS

All protein prompts follow a unified template, where the structured placeholder $\langle$3d protein$\rangle$ is encoded by the structure encoder, `[FASTA]` specifies the protein sequence, and `[description]` provides the textual description. For example:

```
Compose a summary of the protein ⟨3d protein⟩, which is also [FASTA].
[description]
```

To improve robustness, we paraphrase the instruction into multiple variants while keeping the same format (see Table 20 for the full list).

### D.2  MOLECULE ALIGNMENT PROMPTS.

Similar to proteins, all molecule prompts follow a unified template, where the structured placeholder $\langle$3d molecule$\rangle$ is encoded by the structure encoder, `[SMILES]` represents the molecule SMILES string, and `[description]` provides the textual description. For example:

```
Give a breakdown of the molecule ⟨3d molecule⟩, which is also
[SMILES]. [description]
```

To improve robustness, we paraphrase the instruction into multiple variants while keeping the same format (see Table 21 for the full list).

### D.3  COMPLEX ALIGNMENT PROMPTS.

For protein–ligand complexes, all prompts follow a unified template: an instruction applied to the structured placeholder $\langle$3d complex$\rangle$, which internally consists of a protein sequence (FASTA) and a molecule (SMILES). For example:

```
The protein-ligand complex ⟨3d complex⟩ consists of protein [FASTA]
and molecule [SMILES].
```

Here, `[FASTA]` and `[SMILES]` denote the textual placeholders for the protein sequence and molecular string, respectively. To improve robustness, we paraphrase the instruction into multiple variants while keeping the same format (see Table 22 for the full list).

Table 20: Full list of paraphrased protein prompts, where ⟨3d protein⟩ is encoded by the structure encoder, [FASTA] specifies the protein sequence, and [description] provides the textual description.

Give a breakdown of the protein sequence ⟨3d protein⟩, which is also [FASTA]. [description]
Give a breakdown of the FASTA sequence ⟨3d protein⟩, which is also [FASTA]. [description]
Establish an interpretation of the protein sequence ⟨3d protein⟩, which is also [FASTA]. [description]
Establish an interpretation of the FASTA sequenceCreate a representation of the protein sequence's description ⟨3d protein⟩, which is also [FASTA]. [description]
Create a representation of the FASTA sequence's description ⟨3d protein⟩, which is also [FASTA]. [description]
Formulate an explanation of the protein sequence ⟨3d protein⟩, which is also [FASTA]. [description]
Formulate an explanation of the FASTA sequence ⟨3d protein⟩, which is also [FASTA]. [description]
Construct a depiction of the protein sequence ⟨3d protein⟩, which is also [FASTA]. [description]
Construct a depiction of the FASTA sequence ⟨3d protein⟩, which is also [FASTA]. [description]
Form a presentation of the protein sequence ⟨3d protein⟩, which is also [FASTA]. [description]
Form a presentation of the FASTA sequence ⟨3d protein⟩, which is also [FASTA]. [description]
Develop a narrative for the protein sequence ⟨3d protein⟩, which is also [FASTA]. [description]
Develop a narrative for the FASTA sequence ⟨3d protein⟩, which is also [FASTA]. [description]
Prepare a profile of the protein sequence ⟨3d protein⟩, which is also [FASTA]. [description]
Prepare a profile of the FASTA sequence ⟨3d protein⟩, which is also [FASTA]. [description]
Illustrate the characteristics of the protein sequence ⟨3d protein⟩, which is also [FASTA]. [description]
Illustrate the characteristics of the FASTA sequence ⟨3d protein⟩, which is also [FASTA]. [description]
Present a report on the protein sequence ⟨3d protein⟩, which is also [FASTA]. [description]
Present a report on the FASTA sequence ⟨3d protein⟩, which is also [FASTA]. [description]
Generate the description of the protein sequence ⟨3d protein⟩, which is also [FASTA]. [description]
Generate the description of the FASTA sequence ⟨3d protein⟩, which is also [FASTA]. [description]
Offer an analysis of the protein sequence ⟨3d protein⟩, which is also [FASTA]. [description]
Offer an analysis of the FASTA sequence ⟨3d protein⟩, which is also [FASTA]. [description]
Render an explication of the protein sequence ⟨3d protein⟩, which is also [FASTA]. [description]
Render an explication of the FASTA sequence ⟨3d protein⟩, which is also [FASTA]. [description]
Set forth an elucidation of the protein sequence ⟨3d protein⟩, which is also [FASTA]. [description]
Set forth an elucidation of the FASTA sequence ⟨3d protein⟩, which is also [FASTA]. [description]
Compose a summary of the protein sequence ⟨3d protein⟩, which is also [FASTA]. [description]
Compose a summary of the FASTA sequence ⟨3d protein⟩, which is also [FASTA]. [description]
Draw up a delineation of the protein sequence ⟨3d protein⟩, which is also [FASTA]. [description]
Draw up a delineation of the FASTA sequence ⟨3d protein⟩, which is also [FASTA]. [description]
Assemble a sketch of the protein sequence ⟨3d protein⟩, which is also [FASTA]. [description]
Assemble a sketch of the FASTA sequence ⟨3d protein⟩, which is also [FASTA]. [description]
Provide an overview of the protein sequence ⟨3d protein⟩, which is also [FASTA]. [description]
Provide an overview of the FASTA sequence ⟨3d protein⟩, which is also [FASTA]. [description]
Craft an outline of the protein sequence ⟨3d protein⟩, which is also [FASTA]. [description]
Craft an outline of the FASTA sequence ⟨3d protein⟩, which is also [FASTA]. [description]
Produce a detailed account of the protein sequence ⟨3d protein⟩, which is also [FASTA]. [description]
Produce a detailed account of the FASTA sequence ⟨3d protein⟩, which is also [FASTA]. [description]
Build a portrayal of the protein sequence ⟨3d protein⟩, which is also [FASTA]. [description]
Build a portrayal of the FASTA sequence ⟨3d protein⟩, which is also [FASTA]. [description]

## D.4 STRUCTURE-BASED DRUG DESIGN

In the structure-based setting, we design prompts that condition ligand generation on protein binding pockets. Each prompt follows a unified template: an instruction followed by the structured placeholder ⟨3d pocket⟩, which is encoded by the structure encoder. For example:

```
Generate a compound based on the pocket ⟨3d pocket⟩.
```

Notably, these prompts are used in Stage B, where only the pocket features are concatenated with text embeddings. For the ablation in Stage A, the same templates are used with both the placeholder and the keyword "pocket" replaced by "protein," ensuring that generation is conditioned on full protein features rather than pocket features. The full list of paraphrased SBDD prompts is provided in Table 23.

## E USAGE OF LLM

We employed large language models (GPT-5 and GPT-4o) as auxiliary tools during paper writing. Their usage was confined to non-technical writing support, including grammar checking, stylistic adjustments, and improvements in clarity and fluency. All technical ideas, dataset construction, experimental design, and result analysis originate solely from the authors. The use of LLMs did not contribute to the scientific content of this work and served only to facilitate more fluent and polished writing.

Table 21: Full list of paraphrased molecule prompts, where ⟨3d molecule⟩ is encoded by the structure encoder, [SMILES] specifies the molecular representation string, and [description] denotes the textual description.

Give a breakdown of the chemical compound ⟨3d molecule⟩, which is also [SMILES]. [description]
Give a breakdown of the molecule ⟨3d molecule⟩, which is also [SMILES]. [description]
Give a breakdown of the SMILES string ⟨3d molecule⟩, which is also [SMILES]. [description]
Establish an interpretation of the chemical compound ⟨3d molecule⟩, which is also [SMILES]. [description]
Establish an interpretation of the molecule ⟨3d molecule⟩, which is also [SMILES]. [description]
Establish an interpretation of the SMILES stringCreate a representation of the chemical compound's description ⟨3d molecule⟩, which is also [SMILES]. [description]
Create a representation of the molecule's description ⟨3d molecule⟩, which is also [SMILES]. [description]
Create a representation of the SMILES string's description ⟨3d molecule⟩, which is also [SMILES]. [description]
Formulate an explanation of the chemical compound ⟨3d molecule⟩, which is also [SMILES]. [description]
Formulate an explanation of the molecule ⟨3d molecule⟩, which is also [SMILES]. [description]
Formulate an explanation of the SMILES string ⟨3d molecule⟩, which is also [SMILES]. [description]
Construct a depiction of the chemical compound ⟨3d molecule⟩, which is also [SMILES]. [description]
Construct a depiction of the molecule ⟨3d molecule⟩, which is also [SMILES]. [description]
Construct a depiction of the SMILES string ⟨3d molecule⟩, which is also [SMILES]. [description]
Form a presentation of the chemical compound ⟨3d molecule⟩, which is also [SMILES]. [description]
Form a presentation of the molecule ⟨3d molecule⟩, which is also [SMILES]. [description]
Form a presentation of the SMILES string ⟨3d molecule⟩, which is also [SMILES]. [description]
Develop a narrative for the chemical compound ⟨3d molecule⟩, which is also [SMILES]. [description]
Develop a narrative for the molecule ⟨3d molecule⟩, which is also [SMILES]. [description]
Develop a narrative for the SMILES string ⟨3d molecule⟩, which is also [SMILES]. [description]
Prepare a profile of the chemical compound ⟨3d molecule⟩, which is also [SMILES]. [description]
Prepare a profile of the molecule ⟨3d molecule⟩, which is also [SMILES]. [description]
Prepare a profile of the SMILES string ⟨3d molecule⟩, which is also [SMILES]. [description]
Illustrate the characteristics of the chemical compound ⟨3d molecule⟩, which is also [SMILES]. [description]
Illustrate the characteristics of the molecule ⟨3d molecule⟩, which is also [SMILES]. [description]
Illustrate the characteristics of the SMILES string ⟨3d molecule⟩, which is also [SMILES]. [description]
Present a report on the chemical compound ⟨3d molecule⟩, which is also [SMILES]. [description]
Present a report on the molecule ⟨3d molecule⟩, which is also [SMILES]. [description]
Present a report on the SMILES string ⟨3d molecule⟩, which is also [SMILES]. [description]
Generate the description of the chemical compound ⟨3d molecule⟩, which is also [SMILES]. [description]
Generate the description of the molecule ⟨3d molecule⟩, which is also [SMILES]. [description]
Generate the description of the SMILES string ⟨3d molecule⟩, which is also [SMILES]. [description]
Offer an analysis of the chemical compound ⟨3d molecule⟩, which is also [SMILES]. [description]
Offer an analysis of the molecule ⟨3d molecule⟩, which is also [SMILES]. [description]
Offer an analysis of the SMILES string ⟨3d molecule⟩, which is also [SMILES]. [description]
Render an explication of the chemical compound ⟨3d molecule⟩, which is also [SMILES]. [description]
Render an explication of the molecule ⟨3d molecule⟩, which is also [SMILES]. [description]
Render an explication of the SMILES string ⟨3d molecule⟩, which is also [SMILES]. [description]
Set forth an elucidation of the chemical compound ⟨3d molecule⟩, which is also [SMILES]. [description]
Set forth an elucidation of the molecule ⟨3d molecule⟩, which is also [SMILES]. [description]
Set forth an elucidation of the SMILES string ⟨3d molecule⟩, which is also [SMILES]. [description]
Compose a summary of the chemical compound ⟨3d molecule⟩, which is also [SMILES]. [description]
Compose a summary of the molecule ⟨3d molecule⟩, which is also [SMILES]. [description]
Compose a summary of the SMILES string ⟨3d molecule⟩, which is also [SMILES]. [description]
Draw up a delineation of the chemical compound ⟨3d molecule⟩, which is also [SMILES]. [description]
Draw up a delineation of the molecule ⟨3d molecule⟩, which is also [SMILES]. [description]
Draw up a delineation of the SMILES string ⟨3d molecule⟩, which is also [SMILES]. [description]
Assemble a sketch of the chemical compound ⟨3d molecule⟩, which is also [SMILES]. [description]
Assemble a sketch of the molecule ⟨3d molecule⟩, which is also [SMILES]. [description]
Assemble a sketch of the SMILES string ⟨3d molecule⟩, which is also [SMILES]. [description]
Provide an overview of the chemical compound ⟨3d molecule⟩, which is also [SMILES]. [description]
Provide an overview of the molecule ⟨3d molecule⟩, which is also [SMILES]. [description]
Provide an overview of the SMILES string ⟨3d molecule⟩, which is also [SMILES]. [description]
Craft an outline of the chemical compound ⟨3d molecule⟩, which is also [SMILES]. [description]
Craft an outline of the molecule ⟨3d molecule⟩, which is also [SMILES]. [description]
Craft an outline of the SMILES string ⟨3d molecule⟩, which is also [SMILES]. [description]
Produce a detailed account of the chemical compound ⟨3d molecule⟩, which is also [SMILES]. [description]
Produce a detailed account of the molecule ⟨3d molecule⟩, which is also [SMILES]. [description]
Produce a detailed account of the SMILES string ⟨3d molecule⟩, which is also [SMILES]. [description]
Build a portrayal of the chemical compound ⟨3d molecule⟩, which is also [SMILES]. [description]
Build a portrayal of the molecule ⟨3d molecule⟩, which is also [SMILES]. [description]
Build a portrayal of the SMILES string ⟨3d molecule⟩, which is also [SMILES]. [description]

Table 22: Full list of paraphrased complex prompts, where ⟨3d complex⟩ is a structured placeholder rather than a textual input, with [FASTA] specifying the protein sequence and [SMILES] specifying the molecular representation string corresponding to the ligand component.

The protein-ligand complex ⟨3d complex⟩ consists of protein [FASTA] and molecule [SMILES].
The following protein-ligand pair ⟨3d complex⟩ contains a protein [FASTA] and a compound [SMILES].
This complex ⟨3d complex⟩ is formed by protein [FASTA] and ligand [SMILES].
The complex ⟨3d complex⟩ involves a protein sequence [FASTA] and a chemical compound [SMILES].
Here is a protein-ligand complex ⟨3d complex⟩ comprising [FASTA] and [SMILES].
The input complex ⟨3d complex⟩ includes protein [FASTA] and chemical compound [SMILES].
The structure ⟨3d complex⟩ represents a binding between protein [FASTA] and molecule [SMILES].
The biomolecular pair ⟨3d complex⟩ consists of protein [FASTA] and SMILES representation [SMILES].
In this complex ⟨3d complex⟩, a protein [FASTA] interacts with a compound [SMILES].
This protein-ligand pair ⟨3d complex⟩ includes a protein structure [FASTA] and a molecular graph [SMILES].
The following complex ⟨3d complex⟩ illustrates a molecular interaction between [FASTA] and [SMILES].
This protein-ligand complex ⟨3d complex⟩ is composed of protein [FASTA] and chemical entity [SMILES].
In the provided complex ⟨3d complex⟩, the protein [FASTA] is paired with ligand [SMILES].
The example complex ⟨3d complex⟩ is constructed from a protein [FASTA] and molecule [SMILES].
⟨3d complex⟩ is a protein-ligand pair composed of sequence [FASTA] and SMILES [SMILES].
In this molecular protein-ligand complex ⟨3d complex⟩, we observe the interaction between [FASTA] and [SMILES].
The complex ⟨3d complex⟩ showcases a biochemical pair of [FASTA] and [SMILES].
The protein-ligand complex ⟨3d complex⟩ links [FASTA] with [SMILES].
The pair ⟨3d complex⟩ includes a protein [FASTA] and its corresponding ligand [SMILES].
The following structure ⟨3d complex⟩ shows a protein-ligand interaction between [FASTA] and [SMILES].
The complex ⟨3d complex⟩ represents the molecular interaction of sequence [FASTA] and structure [SMILES].
This biomolecular structure ⟨3d complex⟩ is composed of [FASTA] and [SMILES].
⟨3d complex⟩ depicts a protein-ligand binding between protein [FASTA] and molecule [SMILES].
In ⟨3d complex⟩, the protein target [FASTA] is complexed with small molecule [SMILES].
The given molecular complex ⟨3d complex⟩ combines protein [FASTA] and ligand [SMILES].

Table 23: Full list of paraphrased SBDD prompts in Stage B, where ⟨3d pocket⟩ denotes the protein pocket. For the Stage A ablation, both the keyword and placeholder "pocket" are replaced by "protein".

Generate a compound based on the pocket ⟨3d pocket⟩.
Innovate a compound with the pocket ⟨3d pocket⟩ as a foundation.
Assemble a compound in relation to the pocket ⟨3d pocket⟩.
Create a compound influenced by the pocket ⟨3d pocket⟩.
Construct a compound reflecting the essence of the pocket ⟨3d pocket⟩.
Prepare a compound derived from the principles of the pocket ⟨3d pocket⟩.
Innovate a compound in the spirit of the pocket ⟨3d pocket⟩.
Develop a compound that matches the pocket ⟨3d pocket⟩.
Synthesize a compound derived from the pocket ⟨3d pocket⟩.
Manufacture a compound using the pocket ⟨3d pocket⟩ as a basis.
Create a compound that corresponds to the pocket ⟨3d pocket⟩.
Generate a compound that aligns with the pocket ⟨3d pocket⟩.
Synthesize a compound according to the pocket ⟨3d pocket⟩.
Craft a compound in the likeness of the pocket ⟨3d pocket⟩.
Assemble a compound inspired by the essence of the pocket ⟨3d pocket⟩.
Formulate a compound in accordance with the pocket ⟨3d pocket⟩.
Fabricate a compound that adheres to the pocket ⟨3d pocket⟩.
Engineer a compound anchored in the pocket ⟨3d pocket⟩.
Craft a compound that embodies the pocket ⟨3d pocket⟩.
Cultivate a compound with the pocket ⟨3d pocket⟩ in mind.
Design a compound that conforms to the pocket ⟨3d pocket⟩.
Formulate a compound that is influenced by the pocket ⟨3d pocket⟩.
Produce a compound guided by the pocket ⟨3d pocket⟩.
Construct a compound modeled on the pocket ⟨3d pocket⟩.
Design a compound with reference to the pocket ⟨3d pocket⟩.
Generate a compound reflecting the attributes of the pocket ⟨3d pocket⟩.
Produce a compound that incorporates the pocket ⟨3d pocket⟩.
Formulate a compound that mirrors the pocket ⟨3d pocket⟩.
Fabricate a compound utilizing the pocket ⟨3d pocket⟩.
Develop a compound that is rooted in the pocket ⟨3d pocket⟩.
Create a compound that is consistent with the pocket ⟨3d pocket⟩.
Assemble a compound taking the pocket ⟨3d pocket⟩ into account.
Derive a compound from the characteristics of the pocket ⟨3d pocket⟩.
Produce a compound based on the criteria of the pocket ⟨3d pocket⟩.
Compose a compound centered around the pocket ⟨3d pocket⟩.
Fashion a compound in response to the pocket ⟨3d pocket⟩.
Invent a compound informed by the pocket ⟨3d pocket⟩.
Devise a compound inspired by the pocket ⟨3d pocket⟩.
Construct a compound that reflects the pocket ⟨3d pocket⟩.
Design a compound following the pocket ⟨3d pocket⟩.
Develop a compound referencing the pocket ⟨3d pocket⟩.

