# OpenReview forum: "MolChord: Structure–Sequence Alignment for Protein-Guided Drug Design"
_ICLR.cc/2026/Conference — Submitted to ICLR 2026_

### Official Review · Reviewer_MH1k · 2025-10-20

**Soundness:** 2
**Presentation:** 3
**Contribution:** 3
**Rating:** 4
**Confidence:** 4

**Summary:**

The paper proposes **MOLCHORD**, a large-scale framework for structure-based drug design that aligns protein structures with molecular sequences. The method couples a diffusion-based structure encoder (trained on proteins, molecules, and complexes) with a NatureLM-based sequence generator capable of producing SMILES, FASTA, and text. Training proceeds in three stages: (i) alignment pretraining on multimodal structure-to-sequence tasks, (ii) supervised finetuning on protein pocket–ligand complexes, and (iii) reinforcement learning with Direct Preference Optimization (DPO) using docking-based and drug-likeness preference signals. Experiments on the CrossDocked2020 dataset show that MOLCHORD achieves state-of-the-art performance across affinity, drug-likeness, synthesizability, and success rate, while maintaining reasonable diversity. The framework demonstrates improved generalization to unseen proteins, and ablation studies highlight the importance of both the diffusion encoder and the DPO stage. Overall, MOLCHORD advances protein-guided molecular generation by bridging protein structures, sequences, and ligand design within a unified multimodal foundation model.

**Strengths:**

1. **Significant research topic**: The paper tackles a highly impactful problem in structure-based drug design (SBDD), where aligning protein structures and sequences with molecular generation remains a central challenge. Given the immense search space of molecules and the importance of accurate protein–ligand modeling, this is a timely and valuable direction with broad relevance to both the ML and drug discovery communities.
2. **Interesting method**: The proposed MOLCHORD framework is novel in combining a diffusion-based structure encoder with a large autoregressive language model (NatureLM variant), aligned via multimodal pretraining and refined with Direct Preference Optimization (DPO). This design enables flexible integration of protein sequence (FASTA), structure, and pocket-level information with molecular generation, representing an innovative multimodal approach that goes beyond pocket-only baselines.
3. **Good writing and presentation**: The paper is well-structured and clearly written, with a logical flow from motivation to methodology and experiments. The staged training process is explained systematically, and the results are presented with comprehensive baselines and ablations, making the contributions accessible and convincing to the reader.

**Weaknesses:**

1. **Limitation to 1D/2D design**: The method only generates ligands as SMILES strings, without atomic 3D coordinates, meaning it is not a true 3D molecular generator. Therefore, it should also be compared with strong 1D/2D sequence-based baselines such as Reinvent 4 [1], which have been reported to outperform 3D approaches in SBDD [2]. Furthermore, previous 3D baselines (e.g., DecompDiff, MolCRAFT) reported Vina scores on raw generated structures without redocking, whereas MOLCHORD relies only on redocked poses, which may make direct comparisons less straightforward.
2. **Rigid protein assumption**: MOLCHORD assumes proteins are rigid and does not model flexibility or conformational dynamics. Given the importance of induced fit and protein motion in real binding processes, this limitation should be explicitly discussed.
3. **Unclear contribution of protein sequence/global structure**: Although the authors include FASTA and whole-protein features in pretraining and run some prompt-based ablations, there is no quantitative ablation study that isolates how much these inputs improve ligand generation compared to pocket-only conditioning. This weakens the justification for incorporating global protein information.
4. **Unspecified SMILES representation**: The paper does not clarify whether canonical or randomized SMILES are used during training and evaluation. Since SMILES augmentation is widely known to affect generalization and molecular diversity, the choice should be specified and justified.

[1] Reinvent 4: Modern AI–driven generative molecule design.

[2] Structure-based Drug Design Benchmark: Do 3D Methods Really Dominate?

**Questions:**

1. **Choice of DPO vs. direct reward optimization**: Why does the RL stage adopt Direct Preference Optimization (DPO) rather than directly using the original docking and property scores as scalar rewards, especially since those same scores are used for evaluation? Doesn’t the preference formulation risk losing information about score magnitudes?
2. **Role of whole protein structure**: Since the binding pocket is treated as static during ligand design, what is the benefit of incorporating the whole protein structure? How can atoms outside the pocket meaningfully influence the generation process?
3. **Reward design for fused rings**: Why is the number of fused rings explicitly included in the RL reward function, rather than directly using scores of QED and SA?
4. **RL training dynamics**: Could the authors provide a reward curve or training trajectory during the RL process to illustrate stability, convergence, and the trade-off between affinity, diversity, and drug-like properties?

---

> ### Author Response · Authors · 2025-11-21
> **Responses to Reviewer MH1k (Part 1 of 3)**
>
> We sincerely thank the reviewer for the comprehensive and insightful comments, and we have made substantial revisions and new experiments to address all issues as described below.
>
> > W1: Limitation to 1D/2D design: The method only generates ligands as SMILES strings, without atomic 3D coordinates, meaning it is not a true 3D molecular generator. Therefore, it should also be compared with strong 1D/2D sequence-based baselines such as Reinvent 4 [1], which have been reported to outperform 3D approaches in SBDD [2]. Furthermore, previous 3D baselines (e.g., DecompDiff, MolCRAFT) reported Vina scores on raw generated structures without redocking, whereas MOLCHORD relies only on redocked poses, which may make direct comparisons less straightforward.
> >
>
> Our method generates 1D SMILES rather than explicit 3D coordinates. To ensure fair comparison under this setting, as you suggested, we follow the evaluation protocol of “Structure-based Drug Design Benchmark: Do 3D Methods Really Dominate?” (which also includes Reinvent). Specifically, we run MolChord-RL on the same seven targets (1IEP, 3EML, 3NY8, 4RLU, 4UNN, 5MO4, 7L11), using identical generation settings and generating 1000 molecules per target, exactly as described in the benchmark. The top-10 docking results averaged over each target are shown below (see **Appendix Table 12**). We highlight the best and second-best results with **bold** and *italic* text, respectively.:
>
> | **Model** | **1IEP** | **3EML** | **3NY8** | **4RLU** | **4UNN** | **5MO4** | **7L11** | **Avg** |
> | --- | --- | --- | --- | --- | --- | --- | --- | --- |
> | 3DSBDD | -9.05±0.38 | -10.02±0.15 | -10.10±0.24 | -9.80±0.55 | -8.23±0.30 | -8.71±0.45 | -8.47±0.18 | -9.20 |
> | AutoGrow4 | **-13.23±0.11** | **-13.03±0.09** | -11.70±0.00 | -11.20±0.00 | -11.14±0.12 | *-10.38±0.27* | -8.84±0.33 | *-11.36* |
> | Pocket2Mol | -10.17±0.53 | -12.25±0.27 | *-11.89±0.16* | -10.57±0.12 | **-12.20±0.34** | -10.07±0.62 | **-9.74±0.38** | -10.98 |
> | PocketFlow | -12.49±0.70 | -9.25±0.29 | -8.56±0.35 | -9.65±0.25 | -7.90±0.78 | -7.80±0.42 | -8.35±0.31 | -9.14 |
> | ResGen | -10.97±0.29 | -9.25±0.95 | -10.96±0.42 | *-11.75±0.42* | -9.41±0.23 | -10.34±0.39 | -8.74±0.24 | -10.20 |
> | DST | -10.95±0.57 | -10.67±0.24 | -10.54±0.22 | -10.88±0.37 | -9.71±0.19 | -10.03±0.36 | -8.33±0.41 | -10.16 |
> | GraphGA | -10.03±0.41 | -9.89±0.25 | -9.94±0.15 | -10.22±0.39 | -9.32±0.51 | -9.29±0.20 | -7.75±0.32 | -9.49 |
> | MIMOSA | -10.96±0.57 | -10.69±0.24 | -10.51±0.23 | -10.81±0.39 | -9.66±0.25 | -10.02±0.36 | -8.33±0.41 | -10.14 |
> | MolDQN | -6.73±0.12 | -6.51±0.15 | -7.09±0.16 | -6.79±0.26 | -5.92±0.26 | -6.27±0.10 | -6.87±0.20 | -6.60 |
> | Pasithea | -10.86±0.29 | -10.31±0.09 | -10.69±0.27 | -10.92±0.35 | -9.69±0.32 | -9.77±0.21 | -8.06±0.22 | -10.04 |
> | REINVENT | -9.87±0.31 | -9.48±0.39 | -9.61±0.36 | -9.69±0.29 | -8.70±0.25 | -8.92±0.38 | -7.25±0.21 | -9.07 |
> | SCREENING | -10.86±0.26 | -10.90±0.54 | -10.73±0.45 | -10.86±0.22 | -9.80±0.23 | -9.91±0.30 | -8.15±0.26 | -10.17 |
> | SELFIES-VAE-BO | -10.15±0.60 | -9.76±0.12 | -9.99±0.28 | -10.00±0.23 | -9.02±0.33 | -9.18±0.39 | -7.75±0.22 | -9.41 |
> | SMILES GA | -9.56±0.17 | -9.56±0.37 | -10.00±0.26 | -9.61±0.19 | -8.80±0.20 | -9.21±0.23 | -7.54±0.32 | -9.18 |
> | SMILES LSTM HC | -10.38±0.21 | -10.30±0.15 | -10.19±0.12 | -10.49±0.49 | -9.36±0.17 | -9.71±0.43 | -7.90±0.26 | -9.76 |
> | SMILES-VAE-BO | -9.93±0.22 | -9.78±0.10 | -9.96±0.29 | -10.05±0.20 | -9.03±0.30 | -9.18±0.39 | -7.74±0.25 | -9.38 |
> | MolChord-RL (Ours) | *-12.69±0.24* | *-12.39±0.53* | **-13.02±0.23** | **-12.96±0.33** | *-11.38±0.21* | **-11.34±0.18** | *-9.50±0.16* | **-11.90** |
>
> MolChord-RL achieves state-of-the-art results on 3NY8, 4RLU, and 5MO4, and ranks second on the remaining four targets. Importantly, it also obtains the best average performance across all seven targets, demonstrating consistently strong results even when compared directly against Reinvent and the other 1D/2D baselines included in the benchmark.
>
> Regarding the comparison with 3D methods: prior works such as TargetDiff, DecompDiff, and MolCRAFT typically report three affinity-related metrics:
>
> - Vina Score (evaluated on raw generated conformers)
> - Vina Min (after local minimization)
> - Vina Dock (after a full re-docking procedure)
>
> Vina Dock is increasingly used in recent SBDD work—especially for 1D/2D generators—because re-docking eliminates differences in initial conformer generation and yields a consistent evaluation protocol. For fairness, we evaluate all baselines and our model using **Vina Dock** under identical re-docking settings, ensuring comparable and interpretable results.
>
> [1] Reinvent 4: Modern AI–driven generative molecule design.
>
> [2] Structure-based Drug Design Benchmark: Do 3D Methods Really Dominate?

---

> ### Author Response · Authors · 2025-11-21
> **Responses to Reviewer MH1k (Part 2 of 3)**
>
> > W2: Rigid protein assumption: MOLCHORD assumes proteins are rigid and does not model flexibility or conformational dynamics. Given the importance of induced fit and protein motion in real binding processes, this limitation should be explicitly discussed.
> >
>
> We appreciate the reviewer’s comment. MolChord indeed adopts a rigid-protein assumption, which is a current limitation of our framework. We have now added an explicit discussion of this limitation, together with potential extensions to incorporate protein flexibility, in the **Conclusion** section of the revised manuscript.
>
> > W3: Unclear contribution of protein sequence/global structure: Although the authors include FASTA and whole-protein features in pretraining and run some prompt-based ablations, there is no quantitative ablation study that isolates how much these inputs improve ligand generation compared to pocket-only conditioning. This weakens the justification for incorporating global protein information.
> >
> >
> > Q2: **Role of whole protein structure**: Since the binding pocket is treated as static during ligand design, what is the benefit of incorporating the whole protein structure? How can atoms outside the pocket meaningfully influence the generation process?
> >
>
> To directly quantify the contribution of global protein information, we first note that MolChord uses full-protein features only in Stage A (pre-training), while in Stage B (SFT) and Stage C (RL), the sequence generator uses features from the pocket region only. To isolate the effect of global structural information, we conduct an ablation in Stage B where the structure encoder also receives **only the pocket region**, keeping all other components and training settings unchanged. The results are summarized in the table below (see **Lines 1140–1153** in the manuscript).
>
> | **Structure Encoder input** | **Vina Dock($\downarrow$)** | **High Affinity($\uparrow$)** | **QED($\uparrow$)** | **SA($\uparrow$)** | **Lipinski($\uparrow$)** | **Diversity($\uparrow$)** | **Success Rate($\uparrow$)** |
> | --- | --- | --- | --- | --- | --- | --- | --- |
> | Pocket | -7.22 | 45.6% | 0.56 | 0.77 | 4.66 | 0.77 | 25.7% |
> | Full Protein (MolChord) | -7.62 | 55.1% | 0.56 | 0.77 | 4.66 | 0.76 | 33.2% |
>
> We find that removing the whole-protein structure leaves molecular property metrics (QED, SA, Lipinski) and diversity largely unchanged, but causes a clear drop in Vina Dock performance. This shows that global structural context provides binding-relevant cues not captured by the pocket alone, such as overall shape constraints and steric accessibility. In practice, pocket information is sufficient for generating reasonable molecules, but full-protein information leads to better affinity-oriented generation.
>
> > W4: Unspecified SMILES representation: The paper does not clarify whether canonical or randomized SMILES are used during training and evaluation. Since SMILES augmentation is widely known to affect generalization and molecular diversity, the choice should be specified and justified.
> >
>
> We use canonical SMILES throughout both training and evaluation. We do not apply randomized SMILES or other augmentation strategies. This avoids altering the data distribution and to ensure that all sequence variations are learned directly by the model rather than introduced artificially through SMILES-level augmentation. Molecular diversity in our results therefore reflects the model’s generative capacity rather than data augmentation.
>
> > Q1: Choice of DPO vs. direct reward optimization: Why does the RL stage adopt Direct Preference Optimization (DPO) rather than directly using the original docking and property scores as scalar rewards, especially since those same scores are used for evaluation? Doesn’t the preference formulation risk losing information about score magnitudes?
> >
>
> Our decoder is an autoregressive language model, so methods such as PPO or direct reward optimization could in principle be applied. We adopt DPO because of its efficiency. It does not require value-function learning or reward normalization, and in our case we also employ online DPO updates, making it suitable for large-scale sequence generation with docking-based feedback.
>
> Regarding the concern about losing score-magnitude information, we do not observe adverse effects. The preference pairs capture the ranking induced by docking and property scores, allowing the model to substantially improve docking performance while exhibiting only slightly decreases in drug-like properties and diversity. This reflects the expected trade-off rather than a limitation of the preference formulation.

---

> ### Author Response · Authors · 2025-11-21
> **Responses to Reviewer MH1k (Part 3 of 3)**
>
> > Q3: Reward design for fused rings: Why is the number of fused rings explicitly included in the RL reward function, rather than directly using scores of QED and SA?
> >
>
> The key principle behind our approach is to leverage reinforcement learning to target the objectives that are not already well-optimized by the base model. As highlighted in related work like TamGen and MolDiff [1-3], structures with excessive fused rings can be synthetically challenging and often result in unrealistic or undesirable scaffolds, making it essential to include a fused-ring penalty in our reward design. By this, we include the fused-ring count in the RL process.
>
> To validate this design, we also conduct ablations where the reward is replaced by QED-based and SA-based objectives. The corresponding formulations are:
>
> - For Affinity+QED, $R(M, P^{\text{pock}}) = -(S_{\text{Vina}}(M, P^{\text{pock}}) - \lambda_{qed} \cdot QED)$, where  $\lambda_{qed}=$5.
> - For Affinity+SA, $R(M, P^{\text{pock}}) = -(S_{\text{Vina}}(M, P^{\text{pock}}) - \lambda_{sa} \cdot SA)$, where  $\lambda_{sa}=$5.
> - For Affinity+QED+SA, $R(M, P^{\text{pock}}) = -(S_{\text{Vina}}(M, P^{\text{pock}}) - \lambda_{qed} \cdot QED - \lambda_{sa} \cdot SA)$, where  $\lambda_{qed}=5, \lambda_{sa}=$5.
> - For Affinity+QED+SA+FR (fused rings), $R(M, P^{\text{pock}}) = -(S_{\text{Vina}}(M, P^{\text{pock}}) - \lambda_{qed} \cdot QED - \lambda_{sa} \cdot SA + \lambda_{fr} \cdot \max(0, \texttt{FusedRing}(M) - 2))$, where $\lambda_{qed}=5$, $\lambda_{sa}=5$, $\lambda_{fr}=0.5$, and $\texttt{FusedRing}(M)$ denotes the number of fused rings in molecule $M$.
>
>
> In addition, following the reward design used in 3DMolFormer [4] (targeting the **success rate** objective), we perform another DPO ablation using their success-rate reward. All results across these reward variants are summarized in the table below (see also **Appendix Table 18**):
>
> | **Reward Components** | **Vina Dock($\downarrow$)** | **High Affinity($\uparrow$)** | **QED($\uparrow$)** | **SA($\uparrow$)** | **Lipinski($\uparrow$)** | **Diversity($\uparrow$)** | **Success Rate($\uparrow$)** | **Top10 Fused Rings** |
> | --- | --- | --- | --- | --- | --- | --- | --- | --- |
> | Affinity+QED | -8.45 | 72.3% | 0.57 | 0.78 | 4.68 | 0.70 | 49.2% | 1.90 |
> | Affinity+SA | -8.39 | 70.0% | 0.54 | 0.79 | 4.63 | 0.70 | 47.3% | 1.82 |
> | Affinity+QED+SA | -8.45 | 72.0% | 0.57 | 0.80 | 4.73 | 0.71 | 50.3% | 1.87 |
> | Affinity+QED+SA+FR | -8.56 | 74.3% | 0.58 | 0.80 | 4.73 | 0.71 | 53.2% | 1.86 |
> | Follow 3DMolFormer [4] | -8.20 | 66.5% | 0.56 | 0.77 | 4.66 | 0.70 | 43.8% | 1.87 |
> | MolChord-RL (Affinity+FR) | -8.59 | 74.6% | 0.56 | 0.78 | 4.72 | 0.71 | 53.4% | 1.75 |
>
> Across all ablations, we observe that while each reward variant can improve its targeted property, two issues consistently appear: a slight drop in affinity and an increase in fused-ring counts. The combined Affinity + QED + SA + FR reward offers a strong overall balance across affinity, QED, SA, and Lipinski, with the only drawback being a modest rise in fused rings. Overall, both Affinity + QED + SA + FR and our original MolChord-RL (Affinity + FR) represent effective optimization choices. We thank the reviewer for this helpful suggestion.
>
> [1] Wu, Kehan, et al. "TamGen: drug design with target-aware molecule generation through a chemical language model"
>
> [2] Timothy J. Ritchie, et al. "The impact of aromatic ring count on compound developability – are too many aromatic rings a liability in drug design?"
>
> [3] Peng, Xingang, et al. "MolDiff: Addressing the Atom-Bond Inconsistency Problem in 3D Molecule Diffusion Generation"
>
> [4] Hu, Xiuyuan, et al. "3DMolFormer: A Dual-channel Framework for Structure-based Drug Discovery"
>
> > Q4: RL training dynamics: Could the authors provide a reward curve or training trajectory during the RL process to illustrate stability, convergence, and the trade-off between affinity, diversity, and drug-like properties?
> >
>
> We provide the training loss curve of the RL stage, together with the affinity, diversity, and drug-like property metrics evaluated at several representative training steps (0.5, 1, 2, 3 epoch). The table below summarizes the quantitative results, and the corresponding curve figure is included in the appendix (**Section C.1, Figure 6 and Figure 7**).
>
> | **Epochs** | **Vina Dock($\downarrow$)** | **High Affinity($\uparrow$)** | **QED($\uparrow$)** | **SA($\uparrow$)** | **Lipinski($\uparrow$)** | **Diversity($\uparrow$)** | **Success Rate($\uparrow$)** |
> | --- | --- | --- | --- | --- | --- | --- | --- |
> | 0.5 | -8.01 | 62.8% | 0.55 | 0.77 | 4.62 | 0.73 | 39.3% |
> | 1 (MolChord-RL) | -8.59 | 74.6% | 0.56 | 0.78 | 4.72 | 0.71 | 53.4% |
> | 2 | -9.18 | 81.7% | 0.42 | 0.77 | 4.37 | 0.63 | 52.2% |
> | 3 | -9.56 | 86.7% | 0.38 | 0.77 | 4.31 | 0.61 | 53.1% |
>
> The loss curve indicates stable optimization, and the results across training steps clearly illustrate the expected trade-offs among affinity, diversity, and drug-like properties.

---

### Official Review · Reviewer_SxDh · 2025-10-30

**Soundness:** 2
**Presentation:** 2
**Contribution:** 2
**Rating:** 4
**Confidence:** 3

**Summary:**

This paper proposes MOLCHORD to align protein structural representations with molecular representations in SBDD. First, MOLCHORD align protein and molecule structures with their textual descriptions and sequential representations using an autoregressive model called NatureLM. Second, the model is able to guide molecules toward desired properties. To achieve this, this work curates a property-aware dataset by integrating preference data and refine the alignment process using DPO.

**Strengths:**

- MOLCHORD is a unified framework aligning protein, molecule, and text representations
in target-aware molecular design.
- A property-aware dataset is curated for properties guidance and this work uses Direct Preference Optimization (DPO) to refine alignment.

**Weaknesses:**

- Some important SBDD methods are not discussed in this paper, such as [1][2][3]
- It’s better to also report Lipinski metric in Table 1. Also, generation efficiency is also an important factor when evaluating the practically usefulness of the model.
- For the docking metric, it’s better to also report the docking score before performing re-docking to directly evaluate the docking performance of generated molecules.
- CrossDocked2020 is a synthetic dataset and has limited quality for training and evaluating SBDD model. This was also discussed a lot within the community. Could authors elaborate more on the usage of this dataset?


[1] Zhang, Zaixi, and Qi Liu. "Learning subpocket prototypes for generalizable structure-based drug design." ICML\
[2] Zhang, Zaixi, et al. "Molecule generation for target protein binding with structural motifs." ICLR\
[3] Fu, Cong, et al. "Fragment and geometry aware tokenization of molecules for structure-based drug design using language models." ICLR

**Questions:**

Please refer to the weakness part

---

> ### Author Response · Authors · 2025-11-21
> **Responses to Reviewer SxDh (Part 1 of 2)**
>
> We are grateful for the reviewer’s detailed feedback, and we have carefully addressed every concern with additional analyses and clarifications as outlined below.
>
> > W1: Some important SBDD methods are not discussed in this paper, such as [1] [2] [3].
> >
>
> Thanks for pointing the three related work. They address SBDD from different perspectives compared with our method. Specificially, DrugGPS [1] generates 3D ligands by predicting sub-pocket prototypes and assembling atoms around these learned structural motifs. FLAG [2] builds molecules in 3D by sequentially placing fragments according to fragment-level priors and geometric constraints. Frag2Seq [3] uses a geometry- and fragment-aware tokenization to auto-regressively decode fragment sequences into full molecules. In contrast, MolChord employs an autoregressive decoder that directly operates in the SMILES space, offering a complementary approach to SBDD. We have incorporated the three work in **Section C.2.1** in Appendix. Please also kindly refer to our response to “W2” for more detailed comparison.
>
> > W2: It’s better to also report Lipinski metric in Table 1. Also, generation efficiency is also an important factor when evaluating the practically usefulness of the model.
> >
>
> As suggested, we also report **Lipinski** metrics for completeness. For Frag2Seq [3], the paper reports the results of DrugGPS and FLAG, and we include these reported values (except Vina Dock, which is not available) for completeness. The full set of reproduced and reported results is included in **Appendix Table 11** and is also shown below. Note to mention that DrugGPS, FLAG and Frag2Seq mainly report “Vina Score” (evaluated on raw generated conformers) rather than Vina Dock (evaluated after re-docking), so their published results are not directly comparable to our evaluation protocol. Only DrugGPS [1] and FLAG [2] provide publicly available inference outputs, which allows consistent re-docking–based evaluation. We therefore re-evaluate these two methods using the same Vina Dock settings as MolChord for fairness.
>
> | **Methods** | **Vina Dock($\downarrow$)** | **High Affinity($\uparrow$)** | **QED($\uparrow$)** | **SA($\uparrow$)** | **Lipinski($\uparrow$)** | **Diversity($\uparrow$)** | **Success Rate($\uparrow$)** |
> | --- | --- | --- | --- | --- | --- | --- | --- |
> | Reference | -7.45 | - | 0.48 | 0.73 | 4.34 | - | 25.0% |
> | GraphBP | -4.80 | 14.2% | 0.43 | 0.49 | 4.88 | **0.79** | 0.1% |
> | Pocket2Mol | -7.15 | 48.4% | *0.56* | 0.74 | *4.94* | 0.69 | 24.4% |
> | TamGen | -7.48 | 52.6% | *0.56* | 0.77 | 4.88 | 0.75 | 32.4% |
> | TargetDiff | -7.80 | 58.1% | 0.48 | 0.58 | 4.59 | 0.72 | 10.5% |
> | DecompDiff | -8.39 | 64.4% | 0.45 | 0.61 | 4.49 | 0.68 | 24.5% |
> | MolCRAFT | -7.92 | 59.1% | 0.50 | 0.69 | - | 0.72 | 26.8% |
> | FlowSBDD | *-8.50* | 63.4% | 0.47 | 0.51 | - | 0.75 | - |
> | FLAG [2] | -7.06 | 47.8% | 0.49 | 0.70 | 4.66 | 0.70 | 16.9% |
> | DrugGPS [1] | -7.48 | 42.1% | 0.47 | 0.63 | 4.50 | 0.74 | 14.1% |
> | Frag2Seq [3] | - | *65.3%* | **0.65** | 0.64 | **4.99** | 0.71 | - |
> | MolChord | -7.62 | 55.1% | *0.56* | *0.77* | 4.66 | *0.76* | *33.2%* |
> | MolChord-RL | **-8.59** | **74.6%** | *0.56* | **0.78** | 4.72 | 0.71 | **53.4%** |
>
> From the table above, MolChord-RL delivers stronger overall performance than these representative baselines, demonstrating the effectiveness of our approach under a unified evaluation setting.
>
> In addition to performance, generation efficiency is also an important practical factor. Frag2Seq [3] reports the time required to generate 100 molecules per pocket for several methods. Following the same setup, we extend their comparison by adding the runtime of MolChord. The results are shown in the table below (see **Appendix Table 15**):
>
> | **Methods** | **Time (s, $\downarrow$)** |
> | --- | --- |
> | 3D-SBDD | 15986.4 |
> | Pocket2Mol | 2827.3 |
> | GraphBP | 1162.8 |
> | TargetDiff | 3428 |
> | DecompDiff | 6189 |
> | DiffSBDD | 629.9 |
> | FLAG [2] | 1289.1 |
> | DrugGPS [1] | 1007.8 |
> | Lingo3DMol | 1481.9 |
> | Frag2Seq [3] | 48.8 |
> | MolChord | 5.0 |
>
> As shown in the table, most representative SBDD methods require over 1000 seconds to generate 100 molecules per target, whereas our method completes the same workload in 5 seconds. This efficiency is achieved using a single A100 GPU by leveraging batched autoregressive decoding (batch size = 128). The resulting speed highlights one practical advantage of our 1D SMILES generation framework.
>
> [1] Zhang, Zaixi, and Qi Liu. "Learning subpocket prototypes for generalizable structure-based drug design." ICML
>
> [2] Zhang, Zaixi, et al. "Molecule generation for target protein binding with structural motifs." ICLR
>
> [3] Fu, Cong, et al. "Fragment and geometry aware tokenization of molecules for structure-based drug design using language models." ICLR

---

> ### Author Response · Authors · 2025-11-21
> **Responses to Reviewer SxDh (Part 2 of 2)**
>
> > W3: For the docking metric, it’s better to also report the docking score before performing re-docking to directly evaluate the docking performance of generated molecules.
> >
>
> Since the decoder of MolChord is an autoregressive generator that produces only 1D SMILES strings, no 3D coordinates are available at generation time. Therefore, the metric (i.e., docking score before performing re-docking) is not applicable to our method.
>
> For fairness, all models (including ours and 3D-generating baselines) are evaluated under the same protocol during evaluation: every generated molecule is re-docked into the target pocket using identical Vina settings. This ensures that the comparison reflects model quality rather than differences in how initial conformations are produced.
>
> > W4: CrossDocked2020 is a synthetic dataset and has limited quality for training and evaluating SBDD model. This was also discussed a lot within the community. Could authors elaborate more on the usage of this dataset?
> >
>
> We use CrossDocked2020 to ensure fair comparison with prior SBDD models, as it is the standard dataset adopted by most existing methods. Within this constraint, we design a data partitioning strategy that makes more effective use of the available complexes. As described in **Lines 236–244**, we split the complexes into two disjoint subsets, $D_B$ and $D_C$: proteins with more than two associated ligands are assigned to $D_B$ for supervised training, resulting in approximately 95k complexes; the remaining pockets form $D_C$, yielding about 4k complexes for DPO-style preference optimization, consistent with the common practice of separating SFT and alignment data.
>
> Furthermore, as noted in **Lines 272–274**, pockets in $D_C$ are retained only if the model can generate sufficiently diverse candidate molecules (diversity > 0.8). This filtering step results in a final subset of roughly 1k high-quality pockets that provide meaningful preference signals for alignment.
>
> This partitioning strategy enables stable supervision and reliable alignment behavior while remaining fully compliant with the CrossDocked2020 benchmark protocol. The processed $D_B$ and $D_C$ splits will be released publicly to facilitate reproducibility and future research.
>
> Meanwhile, we also evaluated our model on another benchmark, the **SBDD benchmark**, and the results similarly demonstrate the strong capabilities of our approach. For further details, please refer to the response to **Reviewer MH1k** (**W1**).

---

### Official Review · Reviewer_GYXR · 2025-11-01

**Soundness:** 3
**Presentation:** 2
**Contribution:** 3
**Rating:** 6
**Confidence:** 2

**Summary:**

MOLCHORD is a large multimodal model for structure-based drug design (SBDD), aiming to link protein structures with suitable small molecules. In SBDD, finding ligands that bind well to a target protein is essential but difficult because protein 3D structures and molecular properties are hard to align. MOLCHORD solves this by combining a diffusion-based structure encoder that understands 3D geometry with NatureLM, a large autoregressive language model that unifies text, protein sequences (FASTA), and molecular strings (SMILES). Through step-by-step alignment across proteins, molecules, and complexes, followed by supervised fine-tuning and Direct Preference Optimization (DPO), MOLCHORD learns to generate compounds that are drug-like, synthesizable, and high-affinity. Experiments on the CrossDocked2020 dataset show state-of-the-art performance in affinity, QED, synthetic accessibility, and diversity, demonstrating that MOLCHORD is an effective and scalable tool for modern drug discovery.

**Strengths:**

* It presents unified multimodal architecture aligning 3D structures with textual and sequential data (FASTA, SMILES).
* The method integrates DPO-based optimization for controllable molecule generation balancing affinity, drug-likeness, and synthesis feasibility.
* The proposed method outperforms diffusion and graph baselines on CrossDocked2020 while generating realistic, FDA-like molecules.
* It demonstrates computational efficiency suitable for high-throughput virtual screening.

**Weaknesses:**

* Docking-based reward still approximates true binding and ignores ADMET constraints.
* Reproducibility cannot be assessed due to the lack of released code.

**Questions:**

* Is there any plan to release the code, model, or dataset?

---

> ### Author Response · Authors · 2025-11-21
> **Responses to Reviewer GYXR**
>
> We appreciate the reviewer’s constructive suggestions, and we respond to each point in the following paragraphs.
>
> > W1: Docking-based reward still approximates true binding and ignores ADMET constraints.
> >
>
> To investigate the impact of incorporating ADMET-related constraints, we conducted an additional experiment to evaluate the blood-brain barrier (BBB) penetration capability, which is associated with the distribution (i.e., the "D" in ADMET) of molecules. The BBB data is sourced from the **BBB_Martins** dataset [1]. We used **ADMET-AI** [2] as a binary classifier to predict BBB permeability, where molecules with predicted values larger than 0.5 are considered permeable. We test both MolChord and MolChord-RL on 10,000 molecules generated from the CrossDocked2020 test pockets and report the BBBP statistics.
>
> In addition, we modify the DPO reward by combining docking affinity with the BBBP signal. Specifically, we use
>
> $R(M, P^{\text{pock}}) = -(S_{\text{Vina}}(M, P^{\text{pock}}) - \lambda_{bbbp} \cdot BBBP)$,
>
> where $BBBP \in \{0, 1\}$ is the predicted permeability and $\lambda_{bbbp}=2$. We retrain the RL stage using this combined reward and evaluate the resulting molecules under the same protocol. The results are shown below (the first two rows correspond to MolChord and MolChord-RL, and the third row reports the ablation with the combined affinity + BBBP reward):
>
> | **Model** | **Vina Dock($\downarrow$)** | **BBBP** | **QED($\uparrow$)** | **SA($\uparrow$)** | **Lipinski($\uparrow$)** | **Diversity($\uparrow$)** | **Success Rate($\uparrow$)** |
> | --- | --- | --- | --- | --- | --- | --- | --- |
> | MolChord | -7.62 | 0.688 | 0.56 | 0.77 | 4.66 | 0.76 | 33.2% |
> | MolChord-RL | -8.59 | 0.683 | 0.56 | 0.78 | 4.72 | 0.71 | 53.4% |
> | MolChord-RL(Affinity+BBBP) | -8.54 | **0.781** | 0.54 | 0.77 | 4.59 | 0.68 | 49.7% |
>
> We observe that adding the BBBP signal leads to a slight decrease in docking affinity but yields a substantial improvement in BBB penetration, indicating that the combined reward effectively balances the two objectives. These results show that ADMET-aware signals can be integrated into our framework without difficulty and offer a promising direction for extending the model beyond purely docking-based rewards. The corresponding analysis has been added to **Section 4.4** of the manuscript in response to this suggestion.
>
> [1] Ines Filipa Martins, et al. "A Bayesian Approach to in Silico Blood-Brain Barrier Penetration Modeling"
>
> [2] https://github.com/swansonk14/admet_ai
>
> > W2: Reproducibility cannot be assessed due to the lack of released code.
> >
> >
> > Q1: Is there any plan to release the code, model, or dataset?
> >
>
> We will release the full codebase, model checkpoints, and all generated molecules on the test set. The release is currently in preparation and will be made publicly available soon.

---

### Official Review · Reviewer_Z5id · 2025-11-03

**Soundness:** 3
**Presentation:** 3
**Contribution:** 3
**Rating:** 6
**Confidence:** 3

**Summary:**

This paper proposes MOLCHORD, a 4-billion-parameter framework for structure-based drug design (SBDD) that addresses two core challenges in existing methods: poor alignment between protein structural representations and molecular sequence representations, and misalignment between generated molecules and desired pharmacological properties.
MOLCHORD integrates a diffusion-based structure encoder (for capturing 3D geometric features of proteins/molecules) and an autoregressive sequence generator (a variant of NatureLM, for generating SMILES/FASTA/text), with alignment facilitated by a lightweight adapter. It adopts a three-stage training strategy: (1) Adapter pre-training for cross-modal (structure-sequence-text) alignment; (2) Supervised fine-tuning (SFT) on protein-ligand complexes; (3) Direct Preference Optimization (DPO) to refine property alignment.

Experimental results on CrossDocked2020 show that MOLCHORD achieves state-of-the-art (SOTA) performance across key metrics (binding affinity, drug-likeness, synthesizability, diversity) and demonstrates robust out-of-distribution (OOD) generalization.

**Strengths:**

1. Unlike existing methods that rely solely on limited protein-ligand pairs for alignment, MOLCHORD leverages multi-task pre-training (protein-to-FASTA, molecule-to-SMILES, complex-to-text) to unify structural, sequential, and textual representations. This design effectively mitigates the data scarcity of high-quality protein-ligand pairs and enables more robust cross-modal interaction— a key innovation that addresses a long-standing bottleneck in SBDD.

2. The paper curates a property-aware dataset for DPO and introduces a reward function that balances binding affinity (Vina score) with synthetic accessibility (SA) and drug-likeness (via fused ring penalty). This avoids the common trade-off in prior RL-based methods (e.g., BindGPT, MolForm) where affinity is improved at the cost of diversity or drug-likeness.

3. The experimental design is thorough.

**Weaknesses:**

1. The structure encoder is pre-trained on 78M protein structures from AlphaFoldDB/PDB, and the generator on NatureLM’s corpus—but critical pre-training details are missing: How were the 78M protein structures filtered (e.g., sequence identity thresholds, resolution constraints)? Low-quality structures could bias encoder learning. Full hyperparameters for pre-training (e.g., learning rate decay schedule, warm-up steps, batch size adjustment) are not provided, making it impossible for small labs to reproduce the 4.2B-parameter model.

**Questions:**

1. During the DPO dataset curation, the fused ring penalty coefficient ($\lambda$=0.5) is chosen without justification. A sensitivity analysis ($\lambda$=0.2/0.5/0.8) would validate whether this choice is optimal or arbitrary.

---

> ### Author Response · Authors · 2025-11-21
> **Responses to Reviewer Z5id (Part 1 of 2)**
>
> We thank the reviewer for the helpful comments, and we have addressed all points as detailed below.
>
> > W1: The structure encoder is pre-trained on 78M protein structures from AlphaFoldDB/PDB, and the generator on NatureLM’s corpus—but critical pre-training details are missing: How were the 78M protein structures filtered (e.g., sequence identity thresholds, resolution constraints)? Low-quality structures could bias encoder learning. Full hyperparameters for pre-training (e.g., learning rate decay schedule, warm-up steps, batch size adjustment) are not provided, making it impossible for small labs to reproduce the 4.2B-parameter model.
> >
>
> We provide a complete description of the data filtering process and the full set of hyperparameters used during pre-training for both the structure encoder and the sequence generator (see **Appendix Section B** for details):
>
> - Structure Encoder:
>     - Structure dataset construction and filtering: The structure encoder was pre-trained on 78M protein structures derived from both experimentally solved PDB entries and predicted structures from AlphaFoldDB. The exact filtering protocol is as follows:
>         - Experimental PDB structures
>
>             We use the PDB20210930 snapshot and adopt the same quality-control criteria as AlphaFold3: (a) Structures containing more than 300 chains are removed; (b) Structures with resolution worse than 9 Å are discarded; (c) Entries with fewer than 4 amino acids are excluded.
>
>         - Predicted structures from AlphaFoldDB
>         AlphaFoldDB contains more than 200M predicted protein models. To reduce redundancy and ensure structural reliability, we apply: (a) 90% sequence-identity clustering (MMseqs2) and retain only cluster representatives; (b) A minimum global pLDDT threshold of 70 to remove low-confidence predictions.
>         After applying these filters, we obtain approximately 78 million non-redundant, quality-controlled protein structures for encoder pre-training. This ensures that low-quality or noisy structural predictions do not introduce bias into the learned representations.
>     - Full pre-training configuration
>
>         To facilitate reproducibility, we provide the complete set of hyperparameters used to train the 3B-parameter encoder:
>
>         | **Component** | **Configuration** |
>         | --- | --- |
>         | Numerical precision | bfloat16 (bf16) |
>         | Global batch size | 4096 |
>         | Optimizer | AdamW |
>         | Peak learning rate | $1\times10^{-4}$ |
>         | LR schedule | Cosine decay |
>         | Warm-up steps | 2000 |
>         | Total training steps | 200k |
>         | Compute | 128$\times$NVIDIA A100 (80GB) |
>         | Training duration | ~14 days |
>
>         All pre-training follows standard large-scale protein-modeling practices, and we do not rely on proprietary optimization tricks.
>
> - Sequence Generator:
>
>     Following NatureLM, we pre-train the sequence generator in three stages on 64 NVIDIA A100 GPUs over 14 days.
>
>     - Stage 1: training from scratch on 300B SlimPajama tokens with the original LLaMA-3 vocabulary (128,256 tokens), using AdamW with learning rate $3\times10^{-4}$, batch size 4,096, context length 8,192, cosine decay, for 18K steps.
>     - Stage 2: extending the tokenizer with domain-specific scientific tokens (SMILES, FASTA, special modality markers) and training for 4K steps while updating only the new embeddings.
>     - Stage 3: full-model continued pre-training on 80B tokens from mixed-domain corpora, including both interleaved cross-modal data (text–molecule, text–protein, protein–molecule) and single-domain corpora. The training data sources cover C4, PubChem, UniRef90, SwissProt, ZINC, among others. A reduced learning rate of $1\times10^{-4}$ is used for 15K steps.

---

> ### Author Response · Authors · 2025-11-21
> **Responses to Reviewer Z5id (Part 2 of 2)**
>
> > Q1: During the DPO dataset curation, the fused ring penalty coefficient (=0.5) is chosen without justification. A sensitivity analysis (=0.2/0.5/0.8) would validate whether this choice is optimal or arbitrary.
> >
>
> To address the concern regarding the fused-ring penalty coefficient, we performed an extended sensitivity analysis by evaluating $\lambda$ = 0, 0.2, 0.5, 0.8 and 1.0 under the same DPO curation pipeline and with identical training steps. The results are summarized in the table below (Lipinski-related statistics are computed following the implementation in [1]).
>
> | **Penalty($\lambda$)** | **Vina Dock($\downarrow$)** | **High Affinity($\uparrow$)** | **QED($\uparrow$)** | **SA($\uparrow$)** | **Lipinski($\uparrow$)** | **Diversity($\uparrow$)** | **Success Rate($\uparrow$)** | **Top10 Fused Rings** |
> | --- | --- | --- | --- | --- | --- | --- | --- | --- |
> | 0 | -8.72 | 77.7% | 0.53 | 0.77 | 4.57 | 0.67 | 52.7% | 1.99 |
> | 0.2 | -8.74 | 77.4% | 0.52 | 0.78 | 4.57 | 0.67 | 53.2% | 1.92 |
> | 0.5 | -8.59 | 74.6% | 0.56 | 0.78 | 4.72 | 0.71 | 53.4% | 1.75 |
> | 0.8 | -8.47 | 71.9% | 0.55 | 0.78 | 4.62 | 0.71 | 48.9% | 1.61 |
> | 1.0 | -8.37 | 69.6% | 0.53 | 0.77 | 4.57 | 0.70 | 46.2% | 1.71 |
>
> Across this range, the model exhibits broadly stable behavior: smaller coefficients ($\lambda$ = 0 and 0.2) tend to produce better Vina Dock scores but also result in relatively high fused-ring counts. Larger coefficients ($\lambda$ = 0.8 and 1.0) more effectively suppress fused-ring formation but show a decline in several molecular property metrics and vina dock score. Overall, the system does not appear highly sensitive within this range, and multiple settings (e.g., 0.2 or 0.8) perform reasonably well. We select $\lambda$ = 0.5 as a middle-ground choice that maintains a balanced trade-off across docking performance, fused-ring control, molecular properties and diversity (see **Lines 1180–1199** in the manuscript).
>
> [1] Peng, Xingang, et al. "Pocket2Mol: Efficient Molecular Sampling Based on 3D Protein Pockets"

---

### Author Response · Authors · 2025-11-26

Dear Reviewers,

We are very grateful for your thoughtful and constructive feedback. Your insightful comments have significantly contributed to improving the clarity and quality of our work. In response, we have made the following updates:

| **Section** | **Details Added** |
| --- | --- |
| Section 4.4 (Line 473) | **ADMET**-aware reward integration |
| Section 5 (Line 513) | Discussion on limitations (**rigid protein structure**) |
| Appendix B.1 (Line 846) | Data and training details for the **structure encoder** |
| Appendix B.2 (Line 881) | Data and training details for the **sequence decoder** |
| Appendix C.1 (Line 945) | **RL stage** training loss and trade-offs (affinity, diversity, molecule properties) |
| Appendix C.2.1 (Line 1001) | **Complete results** on CrossDocked2020 |
| Appendix C.2.1 (Line 1014) | Results on **SBDD Benchmark** |
| Appendix C.2.1 (Line 1103) | **Efficiency** demonstration |
| Appendix C.2.2 (Line 1140) | Ablation study on **Global Protein Structure** |
| Appendix C.2.2 (Line 1154) | Ablation study on **Reward Components** |
| Appendix C.2.2 (Line 1179) | Ablation study on **Fused-ring Penalty** |

We sincerely appreciate your time and consideration in reviewing our work and look forward to any further discussions.

---

### Author Response · Authors · 2025-11-27
**Kind request for further discussion**

Dear reviewers, AC and SACs,

We sincerely appreciate the time and effort you have dedicated to reviewing our submission. In response to your valuable feedback, we have made several updates and conducted additional experiments to further clarify the contributions of MolChord. We have worked diligently to complete the revisions and welcome any further discussions or the opportunity to provide additional information if needed.

Best regards,

Authors

---

### Author Response · Authors · 2025-12-02
**Summary of Contributions and Responses**

Dear reviewers, AC, and SACs,

We sincerely thank the reviewers for their thoughtful comments and suggestions. While some papers have already gone through the discussion process and received feedback, our submission has not yet undergone this step. We appreciate the opportunity to clarify contributions that may not have been sufficiently highlighted earlier and to address the reviewers' concerns, outlining the revisions we have made in response.

**Core Contributions**:

- Multimodal Alignment: We introduce a novel approach to address a core challenge in protein-to-ligand generation: **mapping proteins to molecular representations**. Instead of relying on limited protein-to-SMILES supervision, our method establishes a more robust alignment by leveraging a structure encoder for three-dimensional representations and a unified generator that jointly models small molecules, proteins, and text. We implement a series of auxiliary **cross-modal** training tasks (e.g., protein-to-FASTA, protein-to-description, molecule-to-SMILES) to strengthen this alignment. Experimental results show that our approach significantly enhances the model’s ability to generate target-specific compounds.
- We partition the CrossDocked2020 training set into two disjoint subsets: $D_B$ for Supervised Fine-Tuning (SFT) and $D_C$ for Direct Preference Optimization (DPO). $D_B$ contains proteins associated with more than two ligands, while $D_C$ includes proteins with one or two ligands. This strategic separation aligns the data with the training objectives: SFT uses the rich, multi-ligand data $D_B$ to teach the model diverse ligand possibilities, while $D_C$ undergoes **property-aware** filtering to select high-quality protein-ligand pairs for DPO, enhancing alignment and refining the generation process for novelty and high binding affinity, preventing the model from simply memorizing training examples. This strategy is helpful to improve the diversity and binding affinity of the generated molecules.

**Main Concerns and Responses:**

- We have explored the integration of **ADMET constraints**, using blood-brain barrier (BBB) penetration as an example. Our results demonstrate that our method has strong potential to improve specific properties beyond docking-based rewards. For further details, please refer to the responses to **Reviewer GYXR**.
- We have provided comprehensive details on the training process for both the structure encoder and the sequence generator, including data filtering protocols and pre-training hyperparameters. These details ensure transparency and reproducibility (see responses to **Reviewer Z5id**).
- We have addressed the concern regarding the usage of the CrossDocked2020 dataset by explaining our property-aware partitioning strategy, which enhances the effectiveness of the dataset for training. Additionally, we introduced more baselines and evaluation metrics, demonstrating not only the superior performance of our approach but also its **remarkable efficiency**. For further details, please refer to the response to **Reviewer SxDh**.
- We compared our model with additional sequence-based methods and demonstrated its capability on an external benchmark (SBDD-benchmark). We also included new ablations on the role of the whole protein structure and reward components, validating our design choices. For further details, please refer to the response to **Reviewer MH1k**.

The contributions and responses outlined above are intended to address all of the reviewers' comments and concerns. We have incorporated the necessary revisions into the revised paper, reflecting the feedback provided. We hope that these clarifications address the reviewers' concerns and provide a clearer understanding of the broader impact and relevance of our contributions.

---

### Meta-Review · Area_Chair_ijvH · 2026-01-06

**Summary:**

MolChord presents a large multimodal framework for protein-guided drug design that aligns protein 3D structure features (diffusion-based encoder) with a unified sequence/text generator (NatureLM variant), and then refines generation using a property-aware DPO stage. Reviewers agree the paper targets an important SBDD problem, and the approach can yield strong CrossDocked2020 results (including improved affinity while maintaining reasonable drug-likeness/synthesizability and competitive efficiency).

Pros
* Tackles an important and timely problem: multimodal modeling for protein-guided SBDD.
* Coherent structure–sequence–text alignment idea; multi-task alignment design is viewed as meaningful.
* Strong empirical results reported after revisions (incl. added baselines/metrics/ablations and efficiency reporting).
* Property-guided refinement via DPO is potentially useful in practice.

Cons
* Key implementation/training details were missing in the original submission, limiting reproducibility and assessment at review time (especially given the model scale and reliance on large pretraining).
* The paper went through major revisions with substantial new experiments/analyses added after initial reviews, which makes it hard to evaluate the original technical contribution and to separate core novelty from the updated empirical results.
* Some limitations remain intrinsic (e.g., reliance on docking-based signals), and open questions about evaluation comparability/positioning versus strong sequence-based SBDD baselines were central to the initial concerns.

The paper originally received a mixed, borderline set of scores, and the final decision is not made lightly. On the positive side, the work addresses an important problem: building a multimodal model for SBD. Reviewers highlighted strong potential in the alignment framework and the (revised) empirical results. However, because critical details were missing in the original submission and the manuscript required major post-review revisions to incorporate essential experiments and clarifications, after carefully weighing the merits and given the competitive nature of ICLR submissions, the current contributions do not outweigh the shortcomings for acceptance this cycle. I encourage the authors to incorporate the feedback more fully and consider resubmitting to a future venue.

**Reviewer Concerns:**

These concerns were partially addressed:
* Z5id: missing pretraining/data filtering details and sensitivity to fused-ring penalty were mostly addressed with added training details and a coefficient sweep.
* SxDh: added Lipinski + efficiency reporting and expanded baseline discussion; clarified CrossDocked2020 usage/partitioning and evaluation protocol rationale.
* GYXR: added an ADMET-style example (BBB) and stated an intent to release artifacts.
* MH1k: added comparisons on an external benchmark, including a strong 1D baseline (Reinvent-style comparison), added ablations on global structure vs pocket input, clarified canonical SMILES choice, and provided RL dynamics/ablations.

**Reviewer Scores:**

* SxDh (4) might move up slightly due to added baselines/metrics, efficiency results, and expanded discussion of dataset/evaluation.
* MH1k (4) might move up slightly given the added external-benchmark comparison, protocol discussion, and targeted ablations.
* Z5id (6) is unlikely to change much (already positive).
* GYXR (6) is Unlikely to change materially; the main remaining concern is broader validity and reproducibility (code release), only partially resolved by the promise/extra BBB experiment.

---

### Decision · Program_Chairs · 2026-01-26

Reject